# A transition phase in late mouse oogenesis impacts DNA methylation of the early embryo

Kristeli Eleftheriou [1], Antonia Peter[1], Ivanna Fedorenko [1], Katy Schmidt[1], Mark Wossidlo [1✉] &
Julia Arand [1]

A well-orchestrated program of oocyte growth and differentiation results in a developmentally competent oocyte. In late oogenesis, germinal vesicle oocytes (GVOs) undergo chromatin remodeling accompanied by transcriptional silencing from an NSN (non-surrounded nucleolus) to an SN (surrounded nucleolus) chromatin state. By analyzing different cytoplasmic and nuclear characteristics, our results indicate that murine NSN-GVOs transition via an intermediate stage into SN-GVOs in vivo. Interestingly, this transition can also be observed ex vivo, including most characteristics seen in vivo, which allows to analyze this transition process in more detail. The nuclear rearrangements during the transition are accompanied by changes in DNA methylation and Tet enzyme-catalyzed DNA modifications. Early parthenogenetic embryos, derived from NSN-GVOs, show lower DNA methylation levels than SN-derived embryos. Together, our data suggest that a successful NSN-SN transition in oogenesis including proper DNA methylation remodeling is important for the establishment of a developmentally competent oocyte for the beginning of life.

[1] Department of Cell and Developmental Biology, Center of Anatomy and Cell Biology, Medical University of Vienna, 1090 Vienna, Austria.
✉email: mark.wossidlo@meduniwien.ac.at

The beginning of mammalian life is a complex undertaking that requires the capability for plasticity for the upcoming changes in cellular potency and first lineage specifications. The foundation for successful fertilization and preimplantation development lies in the transcriptionally silent, mature oocyte that provides the majority of resources (such as stored mRNA and proteins) before the derived embryo activates its own genes for the first time during embryonic genome activation in the maternal-to-zygotic transition (MZT)[1]. A well-orchestrated program of oocyte growth and differentiation is essential to provide a developmentally competent oocyte (see reviewed in[2]).

In late oogenesis, fully grown oocytes, so-called germinal vesicle oocytes (GVOs), undergo processes resulting in the silencing of a formerly transcriptionally active oocyte[3,4]. GVOs are antral oocytes arrested in prophase I of meiosis, which mature to metaphase II oocytes upon hormonal stimulation. Interestingly, when GVOs are isolated from ovaries from type 7–8 follicles (see ref. [5]), they can be classified into different states by their nuclear architecture[6–8]. This classification is especially prominent at the nucleolus, a spherical structure in the nucleus important for ribosomal transcription and assembly[9]. The surrounded nucleolus-type of GVO (SN-GVO) is characterized by the presence of condensed chromatin, which forms a ring-like structure around the nucleolus. The non-surrounded nucleolus-type of GVO (NSN-GVO) is defined by diffuse chromatin in the nucleus, with a few chromatin-dense dot-like regions. Few studies reported a third type, the intermediate-type GVO (INT-GVO), defined by a not yet fully closed dense ring of heterochromatin perinucleolar staining accompanied by a diffuse chromatin staining in the nucleoplasm[6–8]. While NSN-GVOs are transcriptionally active, SN-GVOs do not show nascent transcription[3,4,10]. In the mouse, the three GVO states can be detected in different ratios depending on the age of the female[7]. Noteworthy, NSN- and SN-GVOs have been described in humans and other mammals[11–14]. In addition to changes in nuclear architecture, also differences in the cytoplasm of NSN- and SN-GVOs have been observed, including organelle and microtubule reorganization[6,15,16] and storage of mRNA and proteins. For example, cytoplasmic lattices, as well as the lipid droplet content differ between NSN- and SN-GVOs and can help to distinguish the different types of GVOs[17]. These changes are not only occurring in mice but can also be observed in human oocytes[14].

NSN- and SN-GVOs differ in their meiotic and developmental competencies. Both NSN- and SN-GVOs are capable to resume meiosis and can be fertilized in vitro[10,18]. However, NSN-GVOs display strongly reduced meiotic and developmental competencies[10,19–21], and NSN-derived embryos arrest at the 2-cell stage in mouse preimplantation development, while SN-derived embryos develop to term[10,18,20,21]. Since the discovery of the different types of GVOs, it was unclear whether these types co-exist and progress independently to the germinal vesicle breakdown during oocyte maturation[6], or if they represent transitional states during late oogenesis[6,8,22,23]. Also, it was suggested that SN-configurations represent a step toward atresia[8,24]. Although a pseudo-NSN-SN transition has been observed in mouse oocytes undergoing severe apoptosis under unfavorable conditions[22], a question arising from these studies still stands whether developmentally incompetent NSN-GVOs are capable to transition into SN-GVOs.

During oogenesis, oocytes are subjected to intense epigenetic remodeling, which involves global DNA demethylation in primordial germ cells and the establishment of a new oocyte-specific DNA methylome starting in primary oocytes[25,26]. In GVOs, further dynamic changes in epigenetic marks have been detected between NSN- and SN-GVOs. Here, changes in histone modifications through immunofluorescence have been the primary focus of past studies on GVOs, which observed a spatial reorganization of epigenetic modifications between NSN- and SN-GVOs[7,27]. In general, SN-GVOs showed higher levels of epigenetic modifications than NSN-GVOs[27]. Also, antibody signals for DNA methylation (5-methylcytosine, 5mC), a repressive epigenetic mark depending on the genomic context, show higher levels in SN- compared to NSN-GVOs[27], which correlates with the observed transcriptional silencing of SN-GVOs. The discovery of the Tet enzyme-mediated DNA modifications 5-hydroxymethylcytosine, 5-formylcytosine, and 5-carboxylcytosine (5hmC, 5fC, and 5caC respectively, see ref. [28]) demonstrates that DNA methylation is not a rigid mark that silences genomes, but can be a platform for dynamic epigenome modifications during development. The role of these particular DNA modifications during late oogenesis has not been addressed so far. Moreover, it is not known if different DNA modifications contribute to the developmental competence of GVOs and whether their generation in late oogenesis plays an important role in early embryogenesis.

In this study, we hypothesized that murine GVOs undergo a stepwise transition from NSN- via INT- to SN-GVOs, which can also be achieved ex vivo. Our data show progressive transcriptional silencing and cytoplasmic changes (mitochondrial mass and location, Golgi apparatus remodeling) from NSN- to SN-GVOs. Interestingly, we found that under conditions inhibiting in vitro maturation (IVM) an NSN via INT to SN transition can be observed ex vivo and ex vivo transitioned NSN-derived-SN-GVOs share similar characteristics with SN-GVOs. We further questioned whether this transition of transcriptionally active NSN-GVOs to silent SN-GVOs is accompanied by changes in the epigenome and whether missing the NSN-SN transition leads to abnormal DNA modifications in the early embryo with potentially detrimental consequences. Strikingly, NSN-derived parthenotes, which missed the establishment of an SN-like epigenome, revealed abnormal DNA methylation levels in derived one-cell parthenotes, highlighting the importance of epigenetic remodeling during NSN-SN transition for a successful start of embryogenesis.

## Results

**Mouse GVOs undergo an NSN—INT—SN transition in late oogenesis.** Since it was not clear whether murine NSN-GVOs are capable to transition into SN-GVOs through an intermediate stage, we first aimed to characterize the three types of GVOs (Fig. 1a) in more detail to uncover potential signs of a transitional phase in murine GVOs at the time of collection from ovaries. Similar to previous studies we were able to isolate on average 30% NSN-, 9% INT-, and 61% SN-GVOs from adult mice (Supplementary Table 1, also see refs. [7,22]). One main characteristic of NSN- and SN-GVOs is their transcriptional status, with SN-GVOs being transcriptionally silent[3,4,10]. Performing 5-Ethynyl-uridine (EU) pulse-labeling, we observed that INT-GVOs are characterized by a weak EU signal; indeed suggesting that they represent a transitional state from NSN- to SN-GVOs and that transcriptional activity is progressively decreasing during chromatin remodeling (Fig. 1b, c). In addition, also immunofluorescence for DDX21, a nucleolar RNA helicase, which can be used as a proxy for transcriptional activity of RNA polymerase I and II, supports a continuous transition and gradual transcriptionally silencing from NSN- via INT- to SN-GVOs, with a strong DDX21-signal around the nucleolus of NSN-GVOs, which disappears in INT-GVOs at regions where dense chromatin is starting to be observed (Fig. 1d). In SN-GVOs, where the perinucleolar ring is

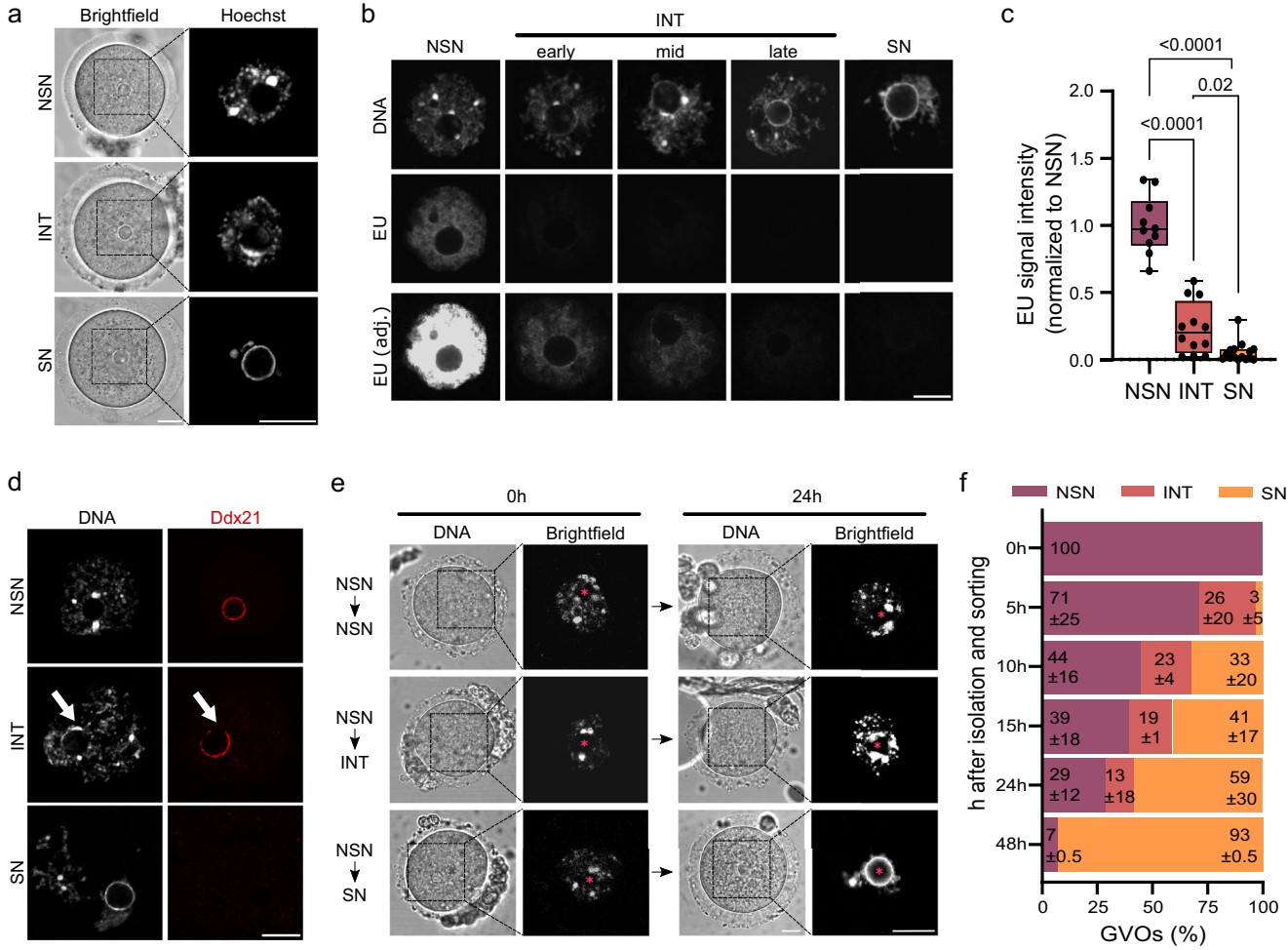

**Fig. 1 Mouse GVOs undergo an NSN-SN transition through an intermediate stage in late oogenesis. a** Representative images of the three types of chromatin configurations in mouse germinal vesicle oocytes (GVOs) stained with Hoechst 33342. Scale bar = 20 μm. **b** Representative images of EU staining of the different types of GVOs. INT-GVOs were further categorized into early, mid, and late depending on the formation of the perinucleolar ring. EU signal intensity was equally adjusted (EU adj.) to compare the EU signals of INT-GVOs. Scale bar = 10 μm. **c** Quantification of EU signal intensity in NSN-, INT-, and SN-GVOs. EU signals were normalized to NSN-GVOs. Each dot represents a single oocyte ($n = 10$ for NSN, $n = 12$ for INT, and $n = 15$ for SN; data are derived from three independent experiments). The box plot is showing the interquartile (box), median (horizontal line), minimum and maximum values (error bars). Statistical significance was calculated using one-way analysis of variance (ANOVA with Tukey's multiple comparison test). **d** Representative immunofluorescence images of the three types of GVOs analyzed for DDX21 ($n > 10$ for each type from two independent experiments). Scale bar = 10 μm. **e** Hoechst 33342 staining of NSN-GVOs 0 h and 24 h after isolation, cultured in IVM media supplemented with IBMX. The asterisk indicates the position of the nucleolus. Scale bar = 20 μm. **f** Quantification of the transition of NSN-GVOs at different timepoints. NSN-GVOs were isolated, sorted, and cultured under maturation inhibiting conditions for 48 h ($n = 42$ NSN-GVOs from two independent experiments).

completely covered with dense chromatin, DDX21-signals were not detected around the nucleolus.

Given this observation of progressive transcriptional silencing from NSN- via INT- to SN-GVOs, we wondered whether NSN-GVOs can transition via an intermediate stage to SN-GVOs ex vivo, as a detailed characterization of this transition is missing. To this end, we analyzed the chromatin configuration of isolated NSN-GVOs at different timepoints cultured for 48 h under maturation inhibiting conditions. We observed that the majority of NSN-GVOs transitioned at 24 h, with 13% at the INT-state and 58.5% at the SN-state (Fig. 1e, f). As we could detect an INT-state for 77% of transitioned SN-GVOs at 24 h, these experiments showed that NSN-GVOs are able to transition ex vivo to SN-GVOs via an intermediate state under IVM inhibiting conditions (Supplementary Fig. 1). At 48 h after GVO isolation, 92.9% of all NSN-GVOs reached the SN-state (Fig. 1f and Supplementary Fig. 1, Clusters II–IV). Interestingly, we observed a large heterogeneity of the NSN-GVO population regarding the

duration of this transition into SN-GVOs (Clusters II–IV, Supplementary Fig. 1). While 23.8% of GVOs start the transitioning earlier than 5 h (Cluster IV), 35.8% start transitioning only after 15 h (Cluster II). Thus, while we can clearly show that NSN-GVOs can transition to SN-GVOs ex vivo, the heterogeneity in the timing of this transition lets us speculate that the population of NSN-GVOs, at the timepoint of isolation from ovaries, exists in differentially advanced states.

**Cytoplasmic changes during the NSN-SN transition in mouse GVOs.** Next, we analyzed morphological and cytoplasmic changes in the NSN- to SN-GVO in vivo transition. When comparing cellular sizes of the three types, we detected a small but significant increase in the oocyte volume in SN-GVOs compared to NSN-GVOs (Fig. 2a), similar to previous observations[6,29]. INT-GVOs are of similar size as NSN-GVOs and are significantly smaller than SN-GVOs (Fig. 2a), indicating

that this volume increase of on average 11% happens after the onset of chromatin remodeling.

We analyzed the three types of GVOs for morphological and cellular differences of selected organelles. Several studies have emphasized the importance of mitochondria in oocyte maturation and development[30–33], which led us to analyze the quantity and activity of mitochondria in the different types of GVOs using live staining with MitoTracker Green and tetramethylrhodamine-

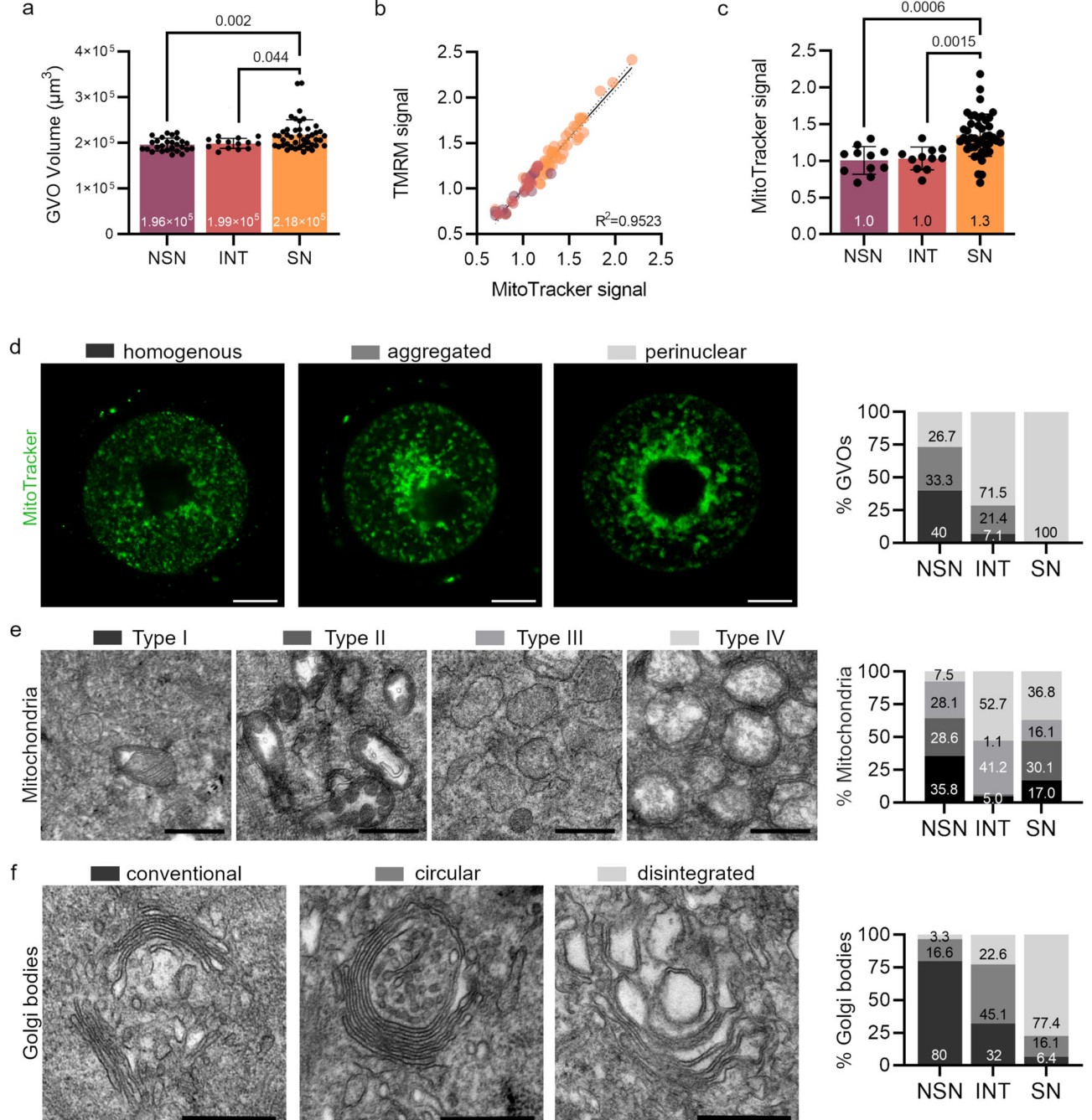

**Fig. 2 Cytoplasmic changes during the NSN-SN transition in murine GVOs. a** Quantification of GVO volume in the different types of GVOs. Each dot represents a single oocyte ($n = 27$ for NSN, $n = 14$ for INT, and $n = 44$ for SN; data are derived from three independent experiments). Statistical significance was calculated using one-way ANOVA with Tukey's multiple comparison test; data are represented as mean ± SD. **b** Correlation of MitoTracker- and TMRM-signal-intensities in GVOs. Each dot represents a single oocyte ($n = 11$ for NSN, $n = 11$ for INT, and $n = 44$ for SN; data are derived from three independent experiments). $R^2$ was calculated using Pearson correlation. **c** Quantification of MitoTracker-signal in NSN-, INT-, and SN-GVOs. Each dot represents a single oocyte ($n = 11$ for NSN, $n = 11$ for INT, and $n = 44$ for SN; data are derived from three independent experiments). Significance was calculated using one-way ANOVA with Tukey's multiple comparison test; data are represented as mean ± SD. **d** Representative images of localization patterns of mitochondria in GVOs and quantification of the different types of localization in NSN-, INT-, and SN-GVOs. Scale bar = 25 μm. **e** Transmission electron microscopy images of different types of mitochondria in GVOs and their distribution in the different GVO types (4 NSN-, 2 INT-, and 4 SN-GVOs were analyzed). Scale bar = 500 nm. **f** Transmission electron microscopy images of different types of the Golgi bodies in GVOs and their distribution in the different GVO types (4 NSN-, 2 INT-, and 4 SN-GVOs were analyzed). Scale bar = 500 nm.

methyl-ester (TMRM) respectively (Supplementary Fig. 2a). We observed a significant increase in mitochondrial mass during the transition from NSN- to SN-GVOs by MitoTracker Green staining (Fig. 2b, c) accompanied by a proportional increase in mitochondrial activity (Fig. 2b and Supplementary Fig. 2b), with INT-GVOs showing similar characteristics as NSN-GVOs. We also detected different mitochondrial localization patterns in the oocyte cytoplasm using MitoTracker Green live staining (Fig. 2d). We classified the observed staining patterns into three categories depending on the mitochondrial distribution in the cytoplasm: aggregation, homogenous, and perinuclear accumulation (Fig. 2d, see ref. 34). In NSN-GVOs, we found a similar distribution of all three patterns across all analyzed oocytes (40% homogenous, 33.3% aggregated, 26.7% perinuclear). In INT-GVOs, the majority represented a perinuclear pattern (71.5%), with 21.4% representing an aggregation and 7.1% homogenous pattern (Fig. 2d). In SN-GVOs, all analyzed GVOs were characterized by the accumulation of mitochondria around the nucleus (perinuclear, Fig. 2d). Transmission electron microscopy showed also different morphological types of mitochondria in the different GVOs which we categorized as Type I–IV (Fig. 2e). Type I mitochondria were mainly found in NSN-GVOs (35.8% of counted mitochondria) and sparsely in INT- and SN-GVOs (5.0% and 17.0% respectively) and were characterized by increased intermembrane space with cristae appearing in a lamellar shape. Mitochondria with ameboid shape and vesicular cristae were classified as Type II and were observed in both NSN- and SN-GVOs (28.6% and 30.1% respectively). As Type III, mitochondria with no cisternae or other internal structures and an electron-dense matrix were classified. These mitochondria were found most prominently in INT- and SN-GVOs (52.7% and 36.8% respectively). Lastly, in Type IV we classified mitochondria with arched cristae and patchy, mostly electron lucid matrix. Type IV mitochondria were the most prominent type found in INT-GVOs (52.7%) and SN-GVOs (36.8%). Even though different types of mitochondria are observed in the different GVO types, the correlation between morphology and activity of mitochondria is still not well understood and deserves future investigations. Notably, we observed dynamic changes in the mitochondrial morphology in the transition from NSN- via INT- to SN-GVOs.

Further analyzing cytoplasmic differences, we also found morphological changes in the Golgi apparatus by transmission electron microscopy (Fig. 2f). During oogenesis, it has been reported that the Golgi apparatus is undergoing ultrastructural changes from flattened stacks in early oogenesis to swollen stacked lamellae and large associated vacuoles in later stages[35]. In the NSN-SN transition, we observed Golgi bodies showing parallel cisternae (defined as conventional) to be most abundant in NSN-GVOs (80%). The number of circular Golgi bodies with parallel cisternae (circular), and disintegrated Golgi bodies were gradually increasing over INT- to SN-GVOs, while conventional Golgi bodies were present at a level of 32% in INT- and only 6.4% in SN-GVOs (Fig. 2f). In SN-GVOs, the disintegrated Golgi apparatus (termed swollen in ref. 35) was the most prominent (77.4%). Moreover, we were not able to detect rough endoplasmic reticulum in analyzed GVOs. Instead, we found smooth endoplasmic reticulum present in all three types of GVOs in different abundancies, with more dilated smooth endoplasmic reticulum in INT-GVOs and hardly detectable in SN-GVOs (Supplementary Fig. 3).

Together, our observed changes in cellular size, as well as quantity, ultrastructural morphology, and localization of vital organelles in the cytoplasm of NSN-, INT-, and SN-GVOs strongly indicate a stepwise transition from developmentally incompetent NSN- to developmentally competent SN-GVOs.

**Mouse GVOs undergo DNA methylation remodeling during the transition from NSN- to SN-GVOs.** Mammalian oocytes undergo massive DNA methylation changes during oogenesis to establish a maternal-specific epigenome competent for early embryogenesis, which starts with a global loss of 5mC in primordial germ cells and a gain of 5mC during the oocyte growth phase[25,26]. Along 5mC, 5hmC levels, catalyzed by Tet enzymes, have also been shown to steadily increase during oogenesis[36]. Tet enzymes are expressed during late oogenesis[36,37] and play a prominent role directly after fertilization in early embryogenesis[37–40]. Thus, we questioned whether Tet enzyme activity can also be observed in the NSN-SN transition in GVOs.

We analyzed the activity of Tet enzymes in the three types of GVOs by immunofluorescence of DNA 5-Cytosine(5 C)-modifications. Interestingly, we observed dynamic changes of 5mC, 5hmC, 5fC, and 5caC from NSN- via INT- to SN-GVOs, indicating a dynamic remodeling of DNA methylation by Tet enzymes in the NSN-SN transition (Fig. 3). The 5mC-signal was steadily increasing during the transition as reported (Fig. 3, see also ref. 27) and was co-localizing with the dense perinucleolar DNA staining in INT- and SN-GVOs, illustrating the progressive increase in pericentric heterochromatin through an intermediate state. 5hmC-signal intensities also increased from NSN- to SN-GVOs, and its spatial distribution changed from being absent around the nucleolus to a weak perinucleolar signal (Fig. 3a, d). Signals for 5mC and 5hmC were only partially co-localizing and 5hmC was mostly concentrated in discrete areas in the nucleoplasm (Fig. 3a). 5fC-signals decreased from NSN- to SN-GVOs, with 5fC becoming visible at the perinucleolar rim in SN-GVOs (Fig. 3b, d). 5caC levels increased similar to 5mC and 5hmC and became increasingly detectable around the nucleolus in SN-GVOs similar to 5fC (Fig. 3c, d). Immunostainings for 5fC and 5caC showed weak signals and their dynamics during the NSN-SN transition suggest that 5fC is metabolized to 5caC or directly replaced with unmodified cytosines, while 5caC is more stable.

The observed increased levels of DNA 5C-modifications at heterochromatic regions around the nucleolus prompted us to investigate whether differences in DNA methylation levels of different repetitive elements can be detected. We analyzed major satellites, which are usually located around the nucleolus[41] and might acquire different DNA modifications during the GVO transition. Furthermore, we chose IAPLTR1a as a hypermethylated representative of retrotransposons, which are characterized by rather high DNA methylation levels during early embryonic development[42–44]. Using hairpin-bisulfite sequencing, we did not detect significant differences in DNA methylation levels and patterns of these specific repeats (Supplementary Fig. 4).

**NSN-derived parthenogenetic one-cell embryos display abnormal DNA methylation.** We wondered whether observed differences in DNA 5C-modifications at the GVO stage will be inherited to the newly developing embryo derived from NSN- or SN-GVOs. Since IVM and in vitro fertilization rates of NSN-GVOs were low (Supplementary Fig. 5a) and obtaining high enough numbers of fertilized NSN-derived embryos was not feasible, and also to exclude potential paternal confounding factors, we in vitro matured NSN- and SN-GVOs to MII oocytes and chemically activated NSN- and SN-derived MII oocytes to generate parthenogenetic one-cell embryos, of which we analyzed levels of DNA 5C-modifications by immunofluorescence.

To this end, we first characterized in vitro maturation and parthenogenetic activation efficiencies from presorted (by Hoechst 33342 staining) NSN- and SN-GVOs. Both, in vitro maturation and parthenogenetic activation, were less efficient in NSN-GVOs

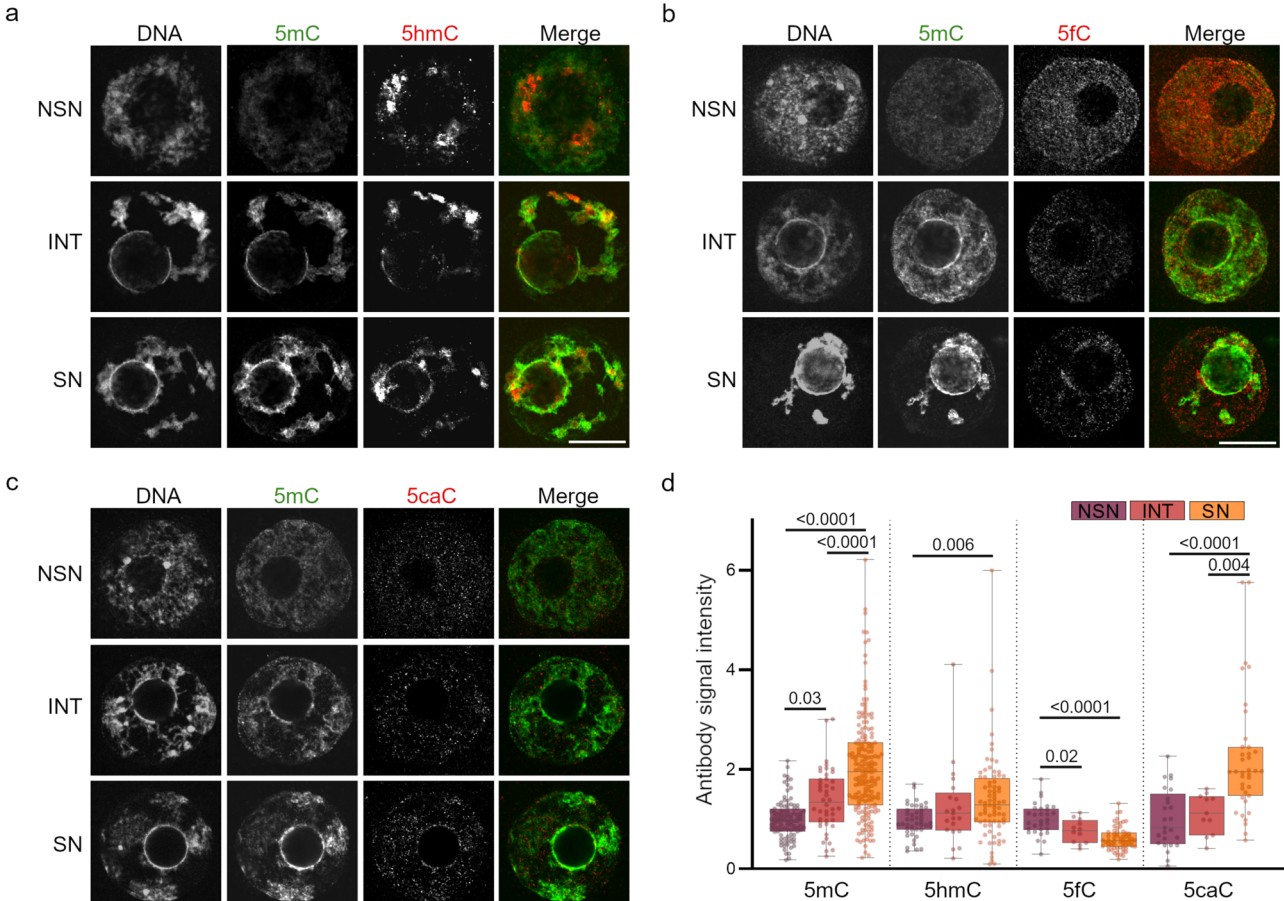

**Fig. 3 Dynamic remodeling of DNA modifications during the NSN-SN transition. a–c** Representative immunofluorescence images of DNA 5C-modifications in NSN-, INT-, and SN-GVOs analyzed for (**a**) 5mC and 5hmC, (**b**) 5mC and 5fC, (**c**) 5mC and 5caC. Scale bar = 15 μm. The fluorescence signal in the merged panels is pseudocoloured. **d** Quantification of 5mC-, 5hmC-, 5fC-, and 5caC-signal intensities in the different GVO stages. Signal intensity was normalized to NSN-GVOs in each experiment. Each dot represents a single oocyte (5mC: $n = 100$ for NSN, $n = 44$ for INT, $n = 178$ for SN, data are derived from 16 independent experiments; 5hmC: $n = 45$ for NSN, $n = 21$ for INT, $n = 76$ for SN, data are derived from 7 independent experiments; 5fC: $n = 30$ for NSN, $n = 12$ for INT, $n = 64$ for SN, data are derived from five independent experiments; 5caC: $n = 25$ for NSN, $n = 11$ for INT, $n = 38$ for SN, data are derived from four independent experiments). The box plots are showing the interquartile (box), median (horizontal line), minimum and maximum values (error bars). Statistical significance was calculated using one-way ANOVA with Tukey's multiple comparison test.

compared to SN-GVOs, as described before (Fig. 4a, Supplementary Fig. 5b, c; see also refs. [10,18,19]). Interestingly, using time-lapse imaging, we noticed a significant delay in the germinal vesicle breakdown and the extrusion of the first polar body, summing up to an on average 8.9 h delay for in vitro maturation of NSN-GVOs compared to SN-GVOs (Fig. 4b). To address whether this could indicate that NSN-GVOs transition to INT- and further to SN-GVOs during IVM before the onset of germinal vesicle breakdown, we microinjected Hoechst-presorted NSN-GVOs or SN-GVOs with mRNA coding for histone H2B-YFP and tracked the changes in chromatin organization through time-lapse microscopy under IVM conditions. We reasoned that tracking a YFP-signal instead of the DNA intercalating Hoechst 33342 dye (monitored by UV light) is less invasive for GVOs during time-lapse imaging. H2B-YFP-signals follow heterochromatin formation during the NSN- to INT- and SN-GVO transition and as such can be used to define NSN- and SN-GVOs similar to Hoechst staining. Out of four *H2B-YFP* mRNA injected NSN-GVOs imaged by time-lapse microscopy, three GVOs did clearly not transition to INT- and SN-GVO state before germinal vesicle breakdown (delay until germinal vesicle breakdown > 9 h), indicating that NSN-GVOs are capable to undergo in vitro maturation directly without transitioning to the INT- or SN-state beforehand (Fig. 4c, Supplementary Movie 1 and 2). Hence, the

timeline of IVM might be useful as a non-invasive marker to discriminate between SN- and NSN-GVOs, as suggested previously[6,21,45], combined with other non-invasive markers such as size, perivitelline space, and position of the germinal vesicle[29,46–48].

We then examined DNA 5C-modification levels of one-cell parthenotes derived from NSN- or SN-GVOs to analyze whether differences in DNA modifications in GVOs will be propagated to the early embryo. Here, we analyzed one-cell parthenotes in G2-phase to study the levels of DNA modifications after the canonical DNA methylation reprogramming phase in one-cell embryos including the Tet enzyme-mediated oxidation and the first replication-dependent passive DNA demethylation. As IVM and parthenogenesis of NSN-GVOs is an inefficient process (Fig. 4a), we limited our analysis to 5mC and 5hmC, as they represent the most prominent DNA 5C-modifications in early preimplantation embryos (see ref. [49]). Strikingly, we detected lower 5mC levels in NSN-derived parthenotes compared to SN-derived parthenotes, suggesting that the missing NSN-SN transition in GVOs leads to persisting lower 5mC levels in NSN-derived parthenotes, which cannot be restored by the canonical DNA methylation reprogramming machinery in one-cell embryos (Fig. 4d, e). In contrast, we did not observe a significant difference between 5hmC-signal intensities between NSN- and SN-derived

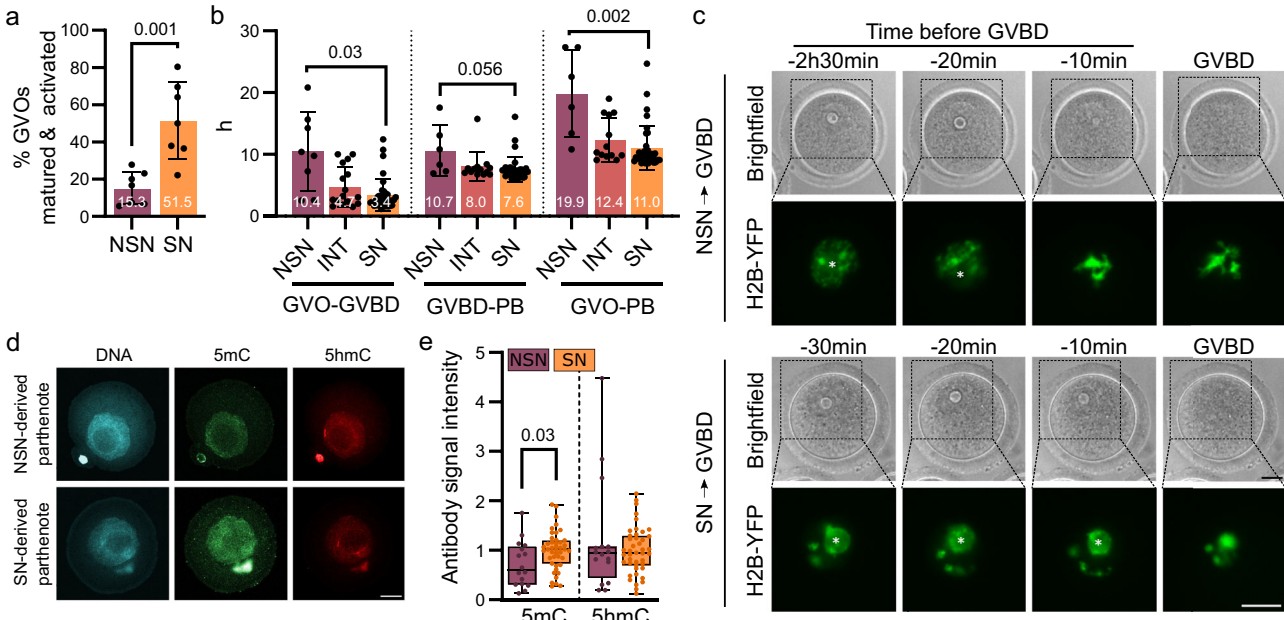

**Fig. 4 NSN-derived parthenogenetic one-cell embryos display abnormal DNA methylation. a** Quantification of the percentage of NSN- and SN-GVOs undergoing in vitro maturation and parthenogenetic activation. Each dot represents one independent experiment (n = 7 for NSN and n = 7 for SN). Statistical significance was calculated using Student's t-test; data are represented as mean ± SD. **b** In vitro maturation timings of NSN- and SN-GVOs. The duration GVOs need for in vitro maturation from start of IVM until germinal vesicle breakdown (GVBD; GVO-GVBD) and from GVBD to extrusion of the first polar body (PB; GVBD-PB) was calculated by time-lapse microscopy. The total time needed for GVOs from start of IVM to extrusion of the polar body is also depicted (GVO-PB). Each dot represents a single oocyte (n = 6 for NSN, n = 13 for INT, n = 34; data are derived from two independent experiments). Statistical significance was calculated using Kruskal–Wallis's test with Dunn's multiple comparisons test; data are represented as mean ± SD. **c** Representative time-lapse images of NSN- and SN-GVOs injected with H2B-YFP undergoing IVM. The asterisk indicates the position of the nucleolus. Scale bar = 20 μm. **d** Representative immunofluorescence images of 5mC and 5hmC DNA modifications in NSN- and SN-derived one-cell parthenotes analyzed at the G2 phase. Scale bar = 20 μm. **e** Quantification of 5mC- and 5hmC-signal intensities in one-cell parthenogenetic embryos derived from NSN- or SN-GVOs. Signal intensities were normalized to SN-GVOs. Each dot represents a single parthenogenetic embryo (n = 16 for NSN and n = 43 for SN; data are derived from five independent experiments). The box plots are showing the interquartile (box), median (horizontal line), minimum and maximum values (error bars). Statistical significance was calculated using Student's t-test.

parthenotes (Fig. 4d, e). This prompts us to speculate that the gained 5mC from NSN- to SN-GVOs is not a substrate for DNA methylation reprogramming in the one-cell stage, which can be confirmed by genomic mapping of 5mC and 5hmC/5fC/5caC. Alternatively, it was reported that NSN-MII oocytes have lower levels of Stella (DPPA3)[50,51], which safeguards the maternal genome from oxidation by Tet enzymes and from de novo methylation[50,52], and could also lead to similar levels of 5hmC in NSN-parthenotes compared to SN-parthenotes – although they start with different 5hmC levels at the GVO state.

In summary, our findings indicate that a missing NSN-SN transition in GVOs results in a hypomethylated maternal methylome in the one-cell stage.

**Ex vivo NSN-derived-SN-GVOs resemble many SN-GVO characteristics**. Using Hoechst staining, we showed that NSN-GVOs transition into SN-GVOs during in vitro culture, thus we wondered to which extent these ex vivo NSN-derived-SN-GVOs (hereby mentioned as N-SN-GVOs) resemble in vivo SN-GVOs. For this, we cultured pre-sorted NSN-GVOs for 24 h or 48 h under in vitro maturation inhibiting conditions and compared derived N-SN-GVOs to in vivo derived SN-GVOs.

Upon a detailed analysis of the chromatin organization of N-SN-GVOs using DNA staining, two types of N-SN-GVOs could be discriminated. Both types showed a fully closed perinucleolar ring; however, while one type closely resembles the in vivo SN-state (type II) with only a few areas with condensed chromatin in the nucleus and absence of diffuse DNA staining, type I showed

diffuse DNA staining all around the nucleus next to the ring-like structure, resembling a mixture of SN- and NSN-chromatin organization (Supplementary Fig. 6a). After 24 h, 15% of transitioned N-SN-GVOs represent type II (typical SN-chromatin state), which increases after 48 h to 60%. This suggests that chromatin condensation during the ex vivo NSN-SN transition takes in general >24 h.

Next, we investigated cytoplasmic characteristics of N-SN-GVOs at 24 h during the transition. Type II N-SN-GVOs were characterized by an oocyte volume close to the volume of SN-GVOs (Supplementary Fig. 6b). Moreover, the mitochondrial distribution of type II N-SN-GVOs resembled the distribution of SN-GVOs, showing a perinuclear localization pattern. Type I N-SN-GVOs resembled the INT-GVO pattern with all three localization patterns observable in different GVOs (Supplementary Fig. 6c).

5mC and 5caC were the most prominently changing DNA 5C-modifications in the NSN-SN transition (Fig. 3d), thus we examined if 5mC and 5caC levels of N-SN-GVOs are changing at 24 h during the transition. N-SN-GVOs showed similar 5mC and 5caC levels as in vivo derived SN-GVOs (Supplementary Fig. 6d), suggesting that N-SN-GVOs undergo similar DNA modification changes ex vivo as NSN-GVOs during in vivo transitioning (Supplementary Fig. 6e).

As N-SN-GVOs resemble many characteristics of SN-GVOs, we wondered if also the meiotic and developmental competence of transitioned N-SN-GVOs is increased compared to NSN-GVOs. For this, we in vitro matured and in vitro fertilized N-SN-GVOs

and tested their developmental competence in early preimplantation development. Upon 24 h transition, germinal vesicle breakdown rates are not increased compared to NSN-GVOs (Supplementary Fig. 7a, c). Interestingly, when performing in vitro fertilization, matured N-SN-GVOs formed zygotes at higher rates than NSN-GVOs (23.1% instead of 4% respectively), but they were not as competent as SN-GVOs (80.2%) and stopped developing at the one-cell stage (Supplementary Fig. 7b, c). Upon 48 h transition, a timepoint at which more N-SN-GVOs resemble type II, we found a clear increase of type II N-SN-GVOs undergoing germinal vesicle breakdown (55.3%) and in derived zygotes (42.9%). Also, the majority of fertilized N-SN-GVOs (48 h) developed until the 2-cell stage (Supplementary Fig. 7c); however, embryos arrested at the 2-cell stage. Thus, we also analyzed the developmental competence of SN-GVOs cultured for 48 h under maturation inhibiting conditions and while these GVOs undergo efficiently in vitro maturation, in vitro fertilization rates dropped (Supplementary Fig. 7c) and they stopped developing at the 2-cell stage, similar to N-SN-GVOs (48 h). Taken together, N-SN-GVOs share characteristics concerning size, mitochondrial localization, and levels of DNA 5C-modifications with in vivo SN-GVOs but their developmental competence is compromised.

## Discussion

During the final step of oogenesis, oocytes acquire nuclear and cytoplasmic maturity in preparation for ovulation. Our results indicate that at this stage in oogenesis NSN-GVOs represent a premature state of GVOs that, upon stimulation, can resume meiosis and can be fertilized, but show an aberrant epigenetic profile in the early embryo.

While previous studies aimed to find potential biomarkers to discriminate NSN- and SN-GVOs in a non-invasive manner[45,47,48], here we profiled mouse GVOs at the stage of isolation of adult ovaries to get a better understanding of their differences and their physiological state. Together our data, including nuclear and cytoplasmic characteristics, clearly reveal a dynamic developmental transition from NSN- via INT- to SN-GVOs. Interestingly, we were able to observe this transition under ex vivo culture conditions. The ex vivo transitioned N-SN-GVOs share similar characteristics with SN-GVOs isolated directly from the ovary, but their developmental competence is compromised, similar to equally long cultured SN-GVOs. We speculate that the transcriptional quiescent state can only be maintained for a specific time to still support developmental competence. Thus, conditions for the ex vivo transition need to be optimized in future experiments (e.g., the timing of inhibition and/or the culture media composition). Importantly, since the observation of an NSN-SN transition in vivo is not feasible, this ex vivo system can provide useful insights into the detailed understanding of this transition in the future and has implications for animal and human reproductive medicine, where novel methods are in development to utilize also any derived premature oocyte for successful fertilization[53].

Our analysis shows that an increase in mitochondrial mass, distinct mitochondrial relocalization, and dynamic changes in the morphology of mitochondria are part of the transition from developmentally incompetent NSN-GVO to competent SN-GVO. It is conceivable that the perinuclear accumulation in SN-GVOs and to a much lesser extent in NSN- and INT-GVOs constitutes a feature of oocyte maturity and that inadequate distribution of mitochondria in NSN-GVOs results in missing chemical energy for the resumption of meiosis, i.e., maturation. Likewise, the Golgi apparatus was reported to undergo structural changes during oogenesis[35]. Our study shows that these morphologically visible changes to a disintegrated Golgi (termed swollen in[35]) actually

happen in late oogenesis during the NSN-SN transition. The restructuring of the Golgi possibly involves the concentration of secretory products necessary for successful early embryonic development.

The changes of DNA modifications observed in this study during the NSN-SN transition reveal an additional phase of DNA methylation remodeling in late oogenesis in the transition to transcriptional silence. Notably, together with the increasing 5mC levels, also 5hmC levels were more prominent in SN-GVOs, suggesting an additional biological function for 5hmC rather than being an intermediate modification for Tet enzyme-mediated active DNA demethylation, similarly to the persisting 5hmC-signals in early preimplantation development[49]. Factors, which bind to Tet enzyme-mediated DNA modifications are still under investigation, and identification of these will help to decipher the biological function of 5hmC in late oogenesis. DNA methylation changes were not detected in selected candidate repetitive sequences (major satellites, IAPLTR1a) indicating that either other genomic loci are undergoing DNA methylation changes or, as bisulfite sequencing cannot distinguish between 5mC/5hmC and 5fC/5caC/C, more specialized sequencing methods are needed to address the dynamic changes of specific DNA modifications at the genomic level. The development of highly specific inhibitors for distinct epigenetic modifiers like Tet enzymes or Dnmts and their precisely timed usage, and also targeted epigenome editing of candidate loci might allow to directly link the contribution of specific epigenetic marks during oogenesis to the developmental competence of the mature oocyte in future experiments.

Failures in establishing an oocyte-specific epigenome in late oogenesis have the potential to be inherited to the early embryo, with potentially detrimental consequences for early life. Our findings revealed that a missing NSN-SN transition in GVOs results in a hypomethylated maternal methylome in the one-cell embryo. Interestingly, it was shown that the DNA methylation status of human oocytes can be used to predict the developmental competence of the embryo[54], suggesting that proper DNA methylation reprogramming during early and late oogenesis is essential for the successful start of life in mammals. Moreover, an elegant study suggested that an NSN nucleus in an SN cytoplasm cannot support efficient blastocyst development[20], highlighting the importance of proper epigenome remodeling in the NSN-SN transition phase.

NSN-derived embryos arrest at the 2-cell stage[10,18,20,21], and it was suggested that impaired $Ca^{2+}$ oscillation in NSN-GVOs is a reason for this lower competence[55]. Here, we show that the transition of NSN- to SN-GVOs is necessary for the establishment of a specific DNA methylome in early embryonic development (Fig. 5). Together with chromatin and cytoplasmic rearrangements, this suggests a pivotal role for the transition in the establishment of a developmentally competent epigenome during embryonic genome activation in derived 2-cell embryos. Interestingly, this transition phase might represent a so far not recognized window of opportunity for the establishment of epimutations that have the potential to impact the beginning of life and possibly future generations and as such should be closely monitored to improve in vitro reproduction in animals and humans.

## Methods

**Animal experiments**. All animal experiments were carried out according to the Austrian Animal Welfare law in agreement with the authorizing committee.

**Collection, classification, and culture of germinal vesicle oocytes**. Fully grown oocytes at the germinal vesicle stage were obtained 48 h post-injection of 5 IU pregnant mare serum gonadotropin (PMSG) from ovaries of 6–20 week old

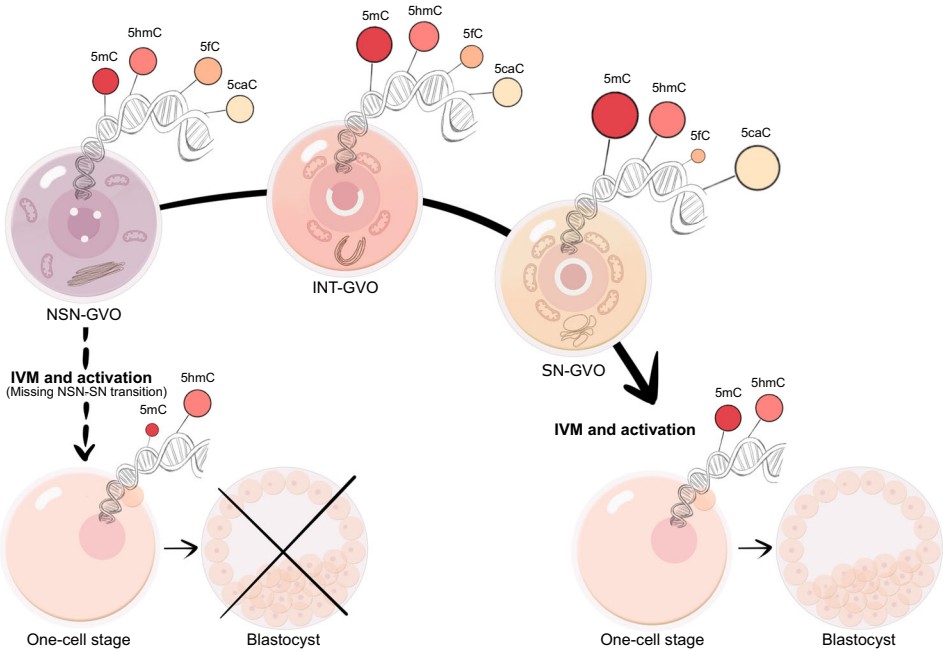

**Fig. 5 Graphical model of dynamic changes of DNA modifications in the NSN-SN transition in late oogenesis.** In late oogenesis, the NSN-SN transition is characterized by dynamic changes in DNA modifications accompanied by nuclear and cytoplasmic rearrangements. A successful transition leads to a developmentally competent embryo. Compromising this transition, the one-cell embryo is characterized by an abnormal epigenetic profile coinciding with developmental arrest, indicating that the NSN-SN transition is necessary for the establishment of a specific epigenome at the beginning of life.

F1(C57BL/6 x DBA) or F1(C57BL/6 x 129SV) female mice and transferred to M2 medium according to standard procedures[56] supplemented with 0.2 mM 3-isobutyl-1-methylxanthine (IBMX; Sigma-Aldrich). To sort for the different GVO states, GVOs were stained for 15 min with 0.1 μg/ml Hoechst 33342 and manually sorted by exposure to UV light for ~1 s using an inverted fluorescence microscope. GVOs, which show diffuse Hoechst staining in the nucleus accompanied by a few dot-like stained denser regions, were categorized as NSN-GVOs; GVOs, which show a partial perinucleolar ring-like structure, as INT-GVOs, and GVOs that show a fully closed perinucleolar ring of Hoechst staining in the nucleus as SN-GVOs (for examples see Fig. 1a). GVOs, which could not be clearly categorized, were excluded from further experiments. NSN-ex vivo-derived SN-GVOs (N-SN-GVOs) were subcategorized into type I and type II, with type I showing a diffuse Hoechst signal all over the nucleoplasm next to the dense perinucleolar ring-like structure, while type II is showing a dense perinucleolar Hoechst signal (Supplementary Fig. 6a). Sorted GVOs were subsequently maintained in α-MEM medium supplemented with 5% FBS, 10 ng/ml EGF and 0.2 mM IBMX in an incubator at 37 °C with 5% $CO_2$ atmosphere. PMSG injection was used for synchronizing the GVOs. Injection of PMSG leads to a slight increase of SN-GVOs and a decrease of NSN-GVOs in adult mice (Supplementary Fig. 8a). In young mice, before the onset of natural SN transitions, the injection of PMSG increases the number of GVOs that can be obtained without inducing the NSN-SN transition (Supplementary Fig. 8a, b). Upon ex vivo transition, there is no difference in the transition rate to the SN-state in juvenile mice with or without PMSG injection (Supplementary Fig. 8c). For in vitro maturation, GVOs were washed in IVM medium (α-MEM medium supplemented with 5% FBS and 10 ng/ml EGF) for 1 h and incubated in a fresh drop of IVM media at 37 °C with 5% $CO_2$ for 16 h to complete meiotic maturation. Oocyte volume ($V_{oocyte}$) was calculated by the following equation (where OD = oocyte diameter):

$$V_{oocyte} = (\pi/6) \times (OD)^3 \quad (1)$$

**GVO microinjections and time-lapse imaging.** After GVO isolation, oocytes were presorted with Hoechst 33342 and co-injected with *H2B-YFP* mRNA and Dextran-tetramethyl-rhodamine (Invitrogen, 3000 MW, 100 μg/ml). *H2B-YFP* mRNA was generated from plasmid DNA using mMESSAGE mMACHINE T7 Ultra Kit (Ambion) and purified using MEGAclear™ Transcription Clean-Up Kit (Ambion). mRNA (100 ng/μL) was injected into GVOs and signal was detected 1–2 h after injection. Microinjected oocytes were cultured in micro-well petri dishes (Vitrolife, Sweden) in IVM medium and monitored using LS560 microscope (Etaluma, San

Diego, US). For time-lapse microscopy, images were taken every 5 min for time-lapse imaging of IVM and every 10-30 min for tracking H2B-YFP during IVM and compiled into a time-lapse movie using ImageJ.

**Oocyte staining and immunofluorescence microscopy.** For immuno-fluorescence analysis, oocytes were briefly washed in M2 medium and zona pellucida was removed by treatment with acidic Tyrode's solution. Oocytes were then fixed with 4% PFA for 20 min at 4 °C, permeabilized with 0.2% TritonX-100 in PBS for 10 min at RT, and then blocked with 0.1% TritonX-100, 1% BSA in PBS overnight at 4°C. Oocytes were immunostained with antibodies against DDX21 (rabbit polyclonal (1:500), Novus Biologicals), anti-5mC (mouse monoclonal (1:100), Eurogentec), anti-5hmC (rabbit polyclonal (1:100), Active Motif), anti-5fC (rabbit polyclonal (1:500), gift from Yi Zhang), anti-5caC (rabbit polyclonal (1:500), Diagenode). To detect DNA modifications (5mC, 5hmC, 5fC, 5caC), oocytes were pretreated with 4 M HCl at RT for 15 min followed by neutralization for 10 min with 100 mM TrisHCl, pH 8.0 and a second fixation. Followed by several washes in blocking solution, oocytes were incubated at RT with anti-mouse or anti-rabbit secondary antibodies for 1.5 h coupled with anti-mouse Alexa Fluor 488 or anti-rabbit Alexa Fluor 568 (10 μg/ml; Molecular Probes). Oocytes were then washed and mounted on slides with a small drop of Vectashield (VectorLab) mounting medium. The oocytes were analyzed on an Olympus Confocal FV3000 microscope. ImageJ software was used to quantify antibody signals of z-stack computed immunofluorescence images (~15 stacks with 0.5 μm per sample, sum projection of z-stacks). For quantification of antibody signals, sum projections of single nuclei/pronuclei were compiled to calculate the corrected total cell fluorescence (CTCF) according to a GitHub protocol by Martin Fitzpatrick (Queensland Brain Institute, University of Queensland, Australia; https://github.com/mfitzp/theolb/blob/master/imaging/measuring-cell-florescence-using-imagej.rst).

**5-Ethynyl-uridine staining.** Nascent RNA staining was performed using the Click-iT nascent RNA detection kit (Life Technologies), following the manufacturer's protocol. Briefly, GVOs were incubated with 10 mM EU for 1 h at 37 °C and 5% $CO_2$. Subsequently, zona pellucida of oocytes was removed with acidic Tyrode's and the oocytes were fixed for 20 min at 4 °C with 4% PFA and permeabilized with 0.2% TritonX-100 in PBS for 10 min at RT. Afterwards, oocytes were incubated with Alexa Fluor 488-azide in Click-iT buffer for 30 min and washed several times with blocking solution.

**Live-staining with MitoTracker Green and TMRM.** For accessing mitochondrial mass and activity of GVOs, live-cell staining with MitoTracker Green FM and TMRM were performed, according to the manufacturer's protocol. MitoTracker Green FM accumulates in the membrane lipids of mitochondria regardless of membrane potential and is useful for evaluating the distribution of mitochondria; TMRM is sequestered by active mitochondria. Oocytes were incubated with 50 nM MitoTracker Green FM (ThermoFisher Scientific), 50 nM TMRM (ThermoFisher Scientific), and 0.1 μg/ml Hoechst 33342 for 30 min in α-MEM medium supplemented with 5% FBS, 10 ng/ml EGF and 0.2 mM IBMX in an incubator at 37 °C with 5% $CO_2$. Oocytes were washed with M2 medium and examined in 10 μl M2 drops supplemented with IBMX under paraffin oil on Olympus Confocal FV3000 microscope.

**Transmission electron microscopy.** Oocytes were preserved by slowly exchanging the supernatant with fixative to a final concentration of paraformaldehyde (2%)/glutaraldehyde (2.5%) / tannic acid (0.1%) in cacodylate buffer (0.1 M). After three washing steps, each followed by briefly centrifuging the samples, oocytes were post-fixed with OsO4 (1%). The fixative was removed by further washing the samples before staining *en bloc* with uranyl acetate (2%) in the dark. Clearing the stain was followed by dehydration in an ascending EtOH series and embedding in Epon resin. Ultrathin sections (50 nm) were collected on formvar-coated slot grids and imaged in an FEI Tecnai20 electron microscope equipped with a 4 K Eagle-CCD camera. Images were processed with Adobe Photoshop Elements.

**Hairpin-bisulfite sequencing.** Hairpin-bisulfite sequencing was performed similarly to Arand et al.[57]. Around 20 NSN- or SN-GVOs per replica were collected and processed to obtain hairpin-bisulfite amplicons for major satellites and IAP_L-TR1a. Briefly, 100 ng salmon sperm DNA was added as carrier DNA and samples were treated with Proteinase K (0.2 mg/ml in 2 mM Tris-HCl, 1 mM EDTA). Proteinase K was inactivated with 8.14 mM Pefabloc SC (Sigma-Aldrich), followed by restriction with DdeI (NEB) for IAP-LTR1a or Eco47I (Fermentas) for major satellites. Subsequently, ligation of the Hairpin linker (Pho-TNACCCGGTADD DDDDDDTACCGGG for IAP_LTR1a and Pho-GACGGGCCTADDDDDDDD TAGGCCC for major Satellites) was performed with T4 Ligase (NEB) and samples were bisulfite treated. Amplification was performed using HOT FIREPol DNA Polymerase (Solis BioDyne) using fusion primer including sequences for preparation of TruSeq amplicon libraries (major Satellites: 95 °C 15' 40x (95 °C 1', 56 °C 2', 72 °C 1') with F-ccatctcatccctgcgtgtctccgacgactacgagtgcgtggaaatttagaaatgtttaatgtag, R-cctatcccctgtgtgccttggcagtcgactacgagtgcgtaacaaaaaaactaaaaatcataaaaa; IAP-LTR1a: 95 °C 15' 45x (95 °C 1', 51 °C 45", 72 °C 1') supplemented with HotStart IT Binding Protein (Affymetrix) with F-ccatctcatccctgcgtgtctccgacgactacgagtgcgtttttttttttaggagagttatattt, R-cctatcccctgtgtgccttggcagtcgactacgagtgcgtat-cactccctaattaactacaac). Amplicons were purified and further amplified with TruSeq primers including barcodes (5 cycles), pooled and sequenced on an Illumina Miseq machine in a 2 × 150 bp run. Obtained reads were analyzed using BiQAnalyzer HT[58]. Data can be found at https://doi.org/10.6084/m9.figshare.21080512.

**In vitro fertilisation and parthenogenetic activation.** Spermatozoa collection and in vitro fertilization procedures were carried out as described[56]. Briefly, sperm was isolated from the *cauda epididymis* of adult F1(C57BL/6×DBA) male mice and capacitated by pre-incubation for 1.5 h in pre-gassed HTF medium. In vitro matured oocytes were placed into HTF medium with capacitated sperm and incubated at 37 °C with 5% $CO_2$ atmosphere. After 4 h embryos were washed in 1:1 HTF:KSOMaa (Sigma-Alrich) for 20 min and further cultured in modified KSO-Maa (supplemented with 3 mg/ml BSA, 5.4 mM glucose). For parthenogenetic activation, in vitro matured oocytes were washed three times with $Ca^{2+}$-free CZB media and incubated for 20 min in $Ca^{2+}$-free CZB media supplemented with 2.5 mM $SrCl_2$ at 37 °C with 5% $CO_2$ atmosphere. After $SrCl_2$ treatment, oocytes were washed three times in KSOMaa medium and cultured in modified KSOMaa medium at 37 °C with 5% $CO_2$ atmosphere. Oocytes were considered activated when each contained one (1PN) or two well-developed pronuclei (2PN). Parthenotes were fixed at 12 h post-activation.

**Statistics and reproducibility.** Statistical analyses were performed using GraphPad Prism software v9 (San Diego, CA USA). A value of $p < 0.05$ was considered statistically significant. Unless otherwise stated in the figure legend, data are shown as the mean ± standard deviation, Student's *t*-test (unpaired) was used for comparisons between two groups, and for multiple comparisons, one-way ANOVA followed by post-hoc Tukey's multiple comparison test was used. The nature of the statistical tests and the *p*-values for the statistical analyses are provided in the associated figure legends wherever applicable. Reproducibility of results was ensured through independent biological replicates, which have been explicitly mentioned wherever applicable. Experiments were performed in at least three independent biological replicates unless otherwise indicated.

## Data availability

The datasets generated during the current study are available from the corresponding author on reasonable request. The underlying data for the hairpin-bisulfite-sequencing can be found at https://doi.org/10.6084/m9.figshare.21080512. The source data for the figures can be found in Supplementary Data 1.

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

## Acknowledgements
We thank Grace Do (Stanford University, USA) for her preliminary work on GVOs and DNA modifications. We thank Yi Zhang (Harvard University, USA) for kindly providing us with 5fC-antibodies, Adelheid Weidinger (Ludwig Boltzmann Institute, Austria) for kindly supporting us with Mitotracker Green FM and TMRM, and Luc Snyers (Medical University of Vienna, Austria) for the H2B-YFP plasmid. We are grateful for support by the Austrian Science Fund (FWF DOI:10.55776/P33984 to M.W.).

## Author contributions
M.W., J.A., and K.E. designed the experiments and interpreted the results. K.E., J.A., A.P., and M.W. performed experiments and analyzed data. K.S. and I.F. generated and analyzed transmission electron microscopy data. M.W., K.E., and J.A. wrote the manuscript and prepared figures with contributions from all authors.

## Competing interests
The authors declare no competing interests.
