## [Peer Review File · Communications Biology]

Reviewers' comments:

Reviewer #1 (Remarks to the Author):

Summary

This study by Eleftheriou et al. examines oocyte epigenomics and organelle states during the key transition to developmental competence in the ultimate stages of oogenesis. Specifically, they meticulously examined the distinction between the non-surrounded nucleolar (NSN) and the surrounded nucleolar (SN) phases, representing developmentally incompetent and competent states of germinal vesicle oocytes (GVOs), respectively. Their work provides novel insights into the dynamic changes of Tet enzyme-mediated DNA modifications, revealing an additional phase of DNA methylation changes in late mouse oogenesis involving oxidation of 5mC by Tet enzymes in the transition from transcriptionally active NSN- to silent SN-GVOs. The authors measured cytosine methylation levels using immunofluorescence, finding that 5hmC levels were more prominent in SN-GVOs, suggesting an additional biological function for 5hmC rather than being an intermediate modification for Tet enzyme-mediated active DNA demethylation, similarly to the persisting 5hmC-signals in early preimplantation development. Intriguingly, they also detected lower levels of 5mC in NSN-derived one-cell parthenogenetic embryos, suggesting that the missing NSN- to SN-transition in NSN-GVOs leads to persisting lower 5mC levels in NSN-derived parthenotes, which cannot be restored by the canonical DNA methylation reprogramming machinery in one-cell embryos. The similar levels of 5hmC found in NSN- and SN-derived parthenotes prompts their speculation that Tet enzymes in one-cell embryos are oxidizing already established 5mC derived from the maternal NSN-epigenome and that the gain of 5mC in SN-GVOs compared to NSN-GVOs is not a substrate for DNA methylation reprogramming in the one-cell embryo, possibly owing to the methylation-stabilizing effects of Stella protein. Taken overall, the findings provide definitive insights into epigenomic changes at a critical phase of oogenesis that underpins the capacity for normal embryonic development. These insights importantly lay the foundation for further experimental interventions that will determine the role of specific epigenomic attributes and modifiers in the acquisition of mammalian oocyte developmental competence.

Comments:

Lines (L)224-225: The exclusion of paternal confounding factors appears to make sense, thus favoring parthenogenetic embryos as the experimental subject to assess developmental competence and fate of GVO epigenomic modifications, but how severe would these paternal confounding factors have been? The authors should justify in greater detail their omission of fertilized embryos as the control for "normal" development and the impact of oocyte epigenomic modifications thereon.

L257-258: Can the authors link the reduced levels of 5mC in NSN-GVOs to developmental competence? The evidence for a causal impact of oocyte epigenetic maturity on developmental competence has not been provided here. (Although previous publications and data presented herein establish the relatively low developmental competence of NSN oocytes)

L543: In Figure 5, the lollipop symbols used to portray the levels of respective modifications, while pleasing to the eye, are not strictly representative of the quantitative changes (as the areas of the circles are proportional to the radii squared, and the larger lollipops thus exaggerate the quantitative differences). Can the authors justify their use of this approach or provide an alternative (such as intensity of shading or a dial showing linear scaling)?

L547-550: The conclusion that "the NSN- to SN-transition is necessary for the establishment of a specific epigenome at the beginning of life" begs the question, how can a requirement for stage-specific oocyte epigenetic modifications be demonstrated experimentally (through perturbation of epigenetic modifications using inhibitors of TET enzymatic function?). While such a direct functional assessment may lie beyond the scope of the present study, delineating the path forward would put the current foundational findings in context and contribute to further work by articulating what is needed

to define the specific role played by each epigenomic modification in oogenesis and potentially in other systems. (The authors provide an example of such a contribution in noting the limitations of current bisulfite sequencing and the need for more specialized methods, L331-333.)

Minor comments:

L 21: "perinuclear" here appears to be a typo, which needs to be corrected to "perinucleolar" to avoid misunderstanding. Also, a greater emphasis on definition of the nucleolus (as a distinct intranuclear organelle) would avoid any such misunderstanding of the location of the perinucleolar chromatin.

L250: It would be helpful to the reader to reiterate here what the "canonical DNA methylation reprogramming phase" consists of.

L265: For clarity, this sentence should be revised to state "...only SN-GVOs being..."

L267: Please correct typo "presume" to "resume"

L269: Advise changing "many" to "previous"

L376: Please specify range of exposure times.

L514: Figure 3 legend should state that the fluorescence signal is pseudocoloured in the merged panels.

Reviewer #2 (Remarks to the Author):

In this paper, Eleftheriou et al. asked if germinal vesicle oocytes (GVOs) with decondensed chromatin (non-surrounded-nucleolus, NSN) transition to GVOs with dense perinuclear chromatin (surrounded nucleolus, SN). This is an interesting and important question in the field. Authors have examined epigenetic, morphological, and cytoplasmic changes in the NSN- to SN-VO transition. Furthermore, DNA methylation was examined in parthenogenetic (PA) embryos derived from NSN-GVOs and SN-GVOs, in order to see whether the germ cell epigenome is transmitted to the next generation. These analyses indicate that SN-GVOs, intermediate-type GVOs (INT-GVOs), and NSN-GVOs show distinct patterns of epigenetic, morphological, and cytoplasmic states. Especially, DNA modification patterns are variable depending on each oocyte stage, and the methylation levels in oocytes affect DNA methylation in 1-cell PA embryos.

This paper confirms the previous findings regarding the differential nuclear and cytoplasmic states between NSN-GVOs and SN-GVOs, further extending their characterization to INT-GVOs. Authors have claimed that NSN-GVOs transition via INT-GVOs (intermediate stage) to SN-GVOs ex vivo. However, this main conclusion is not sufficiently supported by the presented data in this manuscript, as summarized below in this referee's comments. In addition, some of the conclusions are not well supported by the presented data.

Major concerns

1. Authors have claimed that murine NSN-GVOs transition via an intermediate stage into SN-GVOs. This important conclusion is mainly based on the observation that INT-GVOs and SN-GVOs are observed after the culture of NSN-GVOs under the "maturation inhibiting condition". Importantly, NSN-GVOs were selected by Hoechst 33342 staining (Methods), which has been shown to be toxic for development (Versieren, K. et al., 2014 *Zygote*, 22, 32-40). Therefore, these experiments were carried out in a non-physiological condition, the main conclusion has to be confirmed by another experiment that should better represent a physiologically relevant condition. One suggestion is to

trace the transition between NSN-GVOs to SN-GVOs by live cell imaging. Authors have indeed tried a similar experiment in Figure 4C, but unfortunately the resolution of the images is not sufficient to claim the transition from NSN-GVOs to SN-GVOs. Furthermore, the initial classification method for NSN-GVOs, INT-GVOs, and SN-GVOs is extremely important for this paper, but is not well explained in the current manuscript.

2. The definition of INT-GVOs is vague. Authors need to clearly explain the criteria for classifying NSN-GVOs, INT-GVOs, and SN-GVOs. Quantitative statements are ideal for this classification. The difference between NSN-GVOs and INT-GVOs is not clear from Figure 1A.

3. When the developmental competence of NSN-GVOs and SN-GVOs was examined (Figure 4a and 4b), how did you select those oocytes with different categories before being subjected to development tests? Did you use Hoechst 33342 staining? This needs to be explained in the result section.

4. Live cell imaging is a good way forward to answer the main question of this paper. However, the images shown in Figure 4c do not represent the sufficient quality. For example, it is not possible for readers to distinguish NSN-GVOs and SN-GVOs from Figure 4c. Improvement for the live cell experiments is required. Non-biased quantitative approach to assess NSN-GVOs and SN-GVOs would be preferable.

5. Some of the conclusions have to be toned down. For example, to conclude "Our study provides novel insight into the dynamic changes of Tet enzyme-mediated DNA modifications (Line315-316)", knockdown or knockout experiment of Tet in oocytes needs to be performed. To conclude "When this transition is compromised during late oogenesis, the developing preimplantation embryo is characterized by an abnormal epigenetic profile coinciding with developmental arrest" (Line357-359), you need to do more comprehensive studies including fertilized embryos using the NSN-GVOs and SN-GVOs.

6. Regarding the statistical tests with 3 samples, authors have used one-way ANOVA. One-way ANOVA detects significant differences among samples, but not each sample combination. However, P values are shown in each comparison. Can you check and clarify which statistical tests (Multiple Comparison Procedure) were used?

Minor points

7. Line155-173, different types of mitochondria were observed in the different GVO types without strong correlation (Figure 2e). Can you explain more about this in the discussion section?

8. Line458, more explanations are required for statistical methods used for this study.

Reviewer #3 (Remarks to the Author):

The manuscript, "Dynamic changes of DNA modifications in the transition phase of mouse germinal vesicle oocytes impact DNA methylation reprogramming in the early embryo" by Eleftheriou et al compared the differences in ultrastructure, maturation efficiency and epigenetic modification of oocytes with two different nuclear structures, and proved the existence of an intermediate state between NSN- and SN- state. This is an interesting study that advances the understanding of oogenesis, but there are still some questions to be addressed.

1. Oogenesis takes place in the follicle, but oocyte growth is not synchronized with folliculogenesis. Therefore, it is important to clarify which types of follicles (Type 4, Type5, Type 6, or Type 7-8 follicle, refer to: Pedersen T and Peters H., *Reproduction*, 1968; Chen Y et al., *Mol Hum Reprod*, 2020) NSN-GVOs and SN-GVOs exist respectively? At what stage of folliculogenesis does NSN/SN transition occur?

2. In this study, oocytes were collected from ovaries 48h after PMSG injection for analysis. Does PMSG

participate in the regulation of NSN to SN transition? In other words, whether the proportion of NSN-, INT- and SN-GVOs in the population of oocytes changes before and after PMSG injection. This problem can be elucidated by superstimulation in juvenile mice (Postnatal Days 19-20).

3. Fig. 1F showed that NSN-GVOs can be converted to SN-GVOs during in vitro culture. Is the SN-GVOs converted from NSN-GVOs the same as the natural SN-GVOs? That is, do the organelles of the two types of SN-GVOs follow a similar distribution? Do they still differ in size? Are they equally efficient at nuclear maturation? Do they have similar epigenetic modifications and the same potential to develop after fertilization? These questions will determine the practical value of the study and are also of interest to readers.

4. In my opinion, Fig. 4C fails to distinguish the difference in nuclear structure between NSN- and SN-GVOs. What's more, it is difficult to find differences in nuclear maturation between NSN- and SN-GVOs in the movies presented by the authors. If differences exist, then the authors need to make detailed notes for the reader to understand.

5. During the transition from NSN- to SN-GVOs, the morphology and distribution of mitochondria changed significantly. It is suggested that the authors further examine the changes of mtDNA and ATP. Otherwise, the descriptions in line141-143 and line153-154 would lack sufficient support.

OTHER CONCERNS

1. The ABSTRACT is not enough to summarize the research content of the manuscript, please rewrite it.

2. Line 136: Please use volume to measure the change in oocyte size.

3. Line 185: It is not recommended to use "data not shown" to make a vague statement.

4. Line 155-159: No data is presented for the results described

5. Line 161: The authors classify mitochondria into different types and describe the structural characteristics of each type in detail. This is great. However, I believe that readers would be more interested in the functional differences of different mitochondrial types, so I suggest the authors ask experts in mitochondrial research for advice and add descriptions about the functional differences between different mitochondrial types in the manuscript. Similarly, the description of different types of Golgi apparatus should be the same.

6. Fig.4c: The perivitelline space of oocytes in the SN-GVOs group was abnormally large.

7. The DISCUSSION is lengthy and does not succinctly highlight the innovation and importance of the manuscript. In addition, the discussion on epigenetic modification is too detailed, but there is no in-depth study on epigenetic modification in the RESULTS.

Point-by-point response to the reviewers

We thank the reviewers for the exceedingly thorough review and for the helpful and critical comments that allowed us to address the shortcomings of our manuscript. We strongly believe this revision process greatly strengthened our findings and conclusions. We also want to apologize for the long revision time. As probably everybody could experience in the last two years, we were slowed down by cases of Covid-19 of most of the authors during the revision time and more significantly by a shortcoming of mice that could be used for experiments due to a reduction of mouse breeding in our mouse house related to the pandemic.

Please find below our point-by-point response (in blue) to the reviewers (in black). Line number provided in our replies refer to the revised manuscript without tracked changes.

Reviewers' comments:

Reviewer #1 (Remarks to the Author):

Summary

This study by Eleftheriou et al. examines oocyte epigenomics and organelle states during the key transition to developmental competence in the ultimate stages of oogenesis. Specifically, they meticulously examined the distinction between the non-surrounded nucleolar (NSN) and the surrounded nucleolar (SN) phases, representing developmentally incompetent and competent states of germinal vesicle oocytes (GVOs), respectively. Their work provides novel insights into the dynamic changes of Tet enzyme-mediated DNA modifications, revealing an additional phase of DNA methylation changes in late mouse oogenesis involving oxidation of 5mC by Tet enzymes in the transition from transcriptionally active NSN- to silent SN-GVOs. The authors measured cytosine methylation levels using immunofluorescence, finding that 5hmC levels were more prominent in SN-GVOs, suggesting an additional biological function for 5hmC rather than being an intermediate modification for Tet enzyme-mediated active DNA demethylation, similarly to the persisting 5hmC-signals in early preimplantation development. Intriguingly, they also detected lower levels of 5mC in NSN-derived one-cell parthenogenetic embryos, suggesting that the missing NSN- to SN-transition in NSN-GVOs leads to persisting lower 5mC levels in NSN-derived parthenotes, which cannot be restored by the canonical DNA methylation reprogramming machinery in one-cell embryos. The similar levels of 5hmC found in NSN- and SN-derived

parthenotes prompts their speculation that Tet enzymes in one-cell embryos are oxidizing already established 5mC derived from the maternal NSN-epigenome and that the gain of 5mC in SN-GVOs compared to NSN-GVOs is not a substrate for DNA methylation reprogramming in the one-cell embryo, possibly owing to the methylation-stabilizing effects of Stella protein. Taken overall, the findings provide definitive insights into epigenomic changes at a critical phase of oogenesis that underpins the capacity for normal embryonic development. These insights importantly lay the foundation for further experimental interventions that will determine the role of specific epigenomic attributes and modifiers in the acquisition of mammalian oocyte developmental competence.

Comments:

1) Lines (L)224-225: The exclusion of paternal confounding factors appears to make sense, thus favoring parthenogenetic embryos as the experimental subject to assess developmental competence and fate of GVO epigenomic modifications, but how severe would these paternal confounding factors have been? The authors should justify in greater detail their omission of fertilized embryos as the control for "normal" development and the impact of oocyte epigenomic modifications thereon.

We thank the reviewer for this comment, and we agree that the reasoning behind selecting parthenogenetic embryos should be clarified to a better extent in the manuscript. While we

wanted to exclude paternal confounding factors, we also found that obtaining high enough numbers of fertilized embryos after IVM and IVF of NSN-GVOs for downstream analysis is not very feasible, as we tried to derive zygotes from *in vitro* fertilized NSN-matured oocytes (see new Supplementary Fig. S5a). The number of NSN-derived zygotes was so low that

it will take many more experiments and many more mice to reach a significant number of zygotes for this analysis. Unfortunately, this makes such experiments not feasible and cannot

be justified with the 3R rules for animal experiments. As parthenogenetic activation was significantly more efficient for us (see now Supplementary Fig. 5c), we decided to use this approach to study the effect of the NSN-epigenome on the next generation. We now discuss our decision to use parthenotes in more detail in the manuscript and added a new Supplementary Fig. S5a showing the very low IVF rates of NSN-derived mature oocytes.

Lines 220-223: “Since IVM and *in vitro* fertilization (IVF) rates of NSN-GVOs were low (Supplementary Fig. S5a) and obtaining high enough numbers of fertilized NSN-derived embryos was not feasible, and also to exclude potential paternal confounding factors,..”

2) L257-258: Can the authors link the reduced levels of 5mC in NSN-GVOs to developmental competence? The evidence for a causal impact of oocyte epigenetic maturity on developmental competence has not been provided here. (Although previous publications and data presented herein establish the relatively low developmental competence of NSN oocytes).

We agree with the reviewer that this is a very interesting question. Unfortunately, it will be difficult to prove that the DNA methylation status of NSN- or SN-GVOs has a direct causal impact on developmental competence. We could attempt to inhibit Tet enzyme- and Dnmt-activities during the NSN-SN transition; however, current widespread utilized Dnmt- and Tet-inhibitors are known to be rather unspecific and also affect other epigenetic modifiers and key factors (see also answer to reviewer #1 comment 4). A conditional KO of these enzymes would already take effect in the growth phase of oogenesis and a normal epigenome of NSN-GVOs would very likely not be established. We want to mention that an elegant and technically very challenging experiment was performed showing that the transfer of an NSN nucleus to the cytoplasm of an enucleated SN-GVO results in efficient *in vitro* maturation, however, developmental rates to blastocyst are greatly reduced¹. We added this reference to the discussion section (lines 361-363): “Moreover, an elegant study suggested that an NSN nucleus in an SN cytoplasm cannot support efficient blastocyst development¹, highlighting the importance of proper epigenome remodeling in the NSN-SN transition phase.” While this experiment strongly points towards the importance of the “maturity” of the chromatin for preimplantation development, we do not want to claim that DNA methylation is the only prerequisite for developmental competence. We discuss this better now in the main text (lines 365-369): “Here, we show that the transition of NSN- to SN-GVOs is necessary for the

establishment of a specific DNA methylome in early embryonic development (Fig. 5). Together with chromatin and cytoplasmic rearrangements, this suggests a pivotal role for the NSN-SN transition in the establishment of a developmentally competent epigenome during embryonic genome activation in derived 2-cell embryos.”

3) L543: In Figure 5, the lollipop symbols used to portray the levels of respective modifications, while pleasing to the eye, are not strictly representative of the quantitative changes (as the areas of the circles are proportional to the radii squared, and the larger lollipops thus exaggerate the quantitative differences). Can the authors justify their use of this approach or provide an alternative (such as intensity of shading or a dial showing linear scaling)?

We thank the reviewer for this comment, and we agree that it is not strictly representative of the quantitative changes. We have adjusted the figure so the quantitative changes in the DNA modifications are accurately reflected in the diameter of the circles.

4) L547-550: The conclusion that “the NSN- to SN-transition is necessary for the establishment of a specific epigenome at the beginning of life” begs the question, how can a requirement for stage-specific oocyte epigenetic modifications be demonstrated experimentally (through perturbation of epigenetic modifications using inhibitors of TET enzymatic function?). While such a direct functional assessment may lie beyond the scope of the present study, delineating the path forward would put the current foundational findings in context and contribute to further work by articulating what is needed to define the specific role played by each epigenomic modification in oogenesis and potentially in other systems.

(The authors provide an example of such a contribution in noting the limitations of current bisulfite sequencing and the need for more specialized methods, L331-333.)

We are thankful to the reviewer for this comment and added a related discussion point in the main text: (lines 350-354) “The development of highly specific inhibitors for distinct epigenetic modifiers like Tet enzymes or Dnmts and their precisely timed usage, and also targeted epigenome editing for specific candidate loci might allow to directly link the contribution of specific epigenetic marks during oogenesis to the developmental competence of the mature oocyte in future experiments.”

Minor comments:

L21: “perinuclear” here appears to be a typo, which needs to be corrected to "perinucleolar" to avoid misunderstanding. Also, a greater emphasis on definition of the nucleolus (as a distinct intranuclear organelle) would avoid any such misunderstanding of the location of the perinucleolar chromatin.

We thank the reviewer for noticing. We are now more careful avoiding the typo “perinuclear” and also now give a short description of the nucleolus in the introduction (lines 42-45): “...they can be classified into different states by their nuclear architecture²⁻⁴. This classification is especially prominent at the nucleolus, a spherical structure in the nucleus important for ribosomal transcription and assembly⁵.”

L250: It would be helpful to the reader to reiterate here what the "canonical DNA methylation reprogramming phase" consists of.

We agree and now provide more details on what we mean by this phrase; see (lines 249-251) “...after the canonical DNA methylation reprogramming phase in one-cell embryos, including the Tet enzyme-mediated oxidation and the first replication-dependent passive DNA demethylation”.

L265: For clarity, this sentence should be revised to state “...only SN-GVOs being...”

This sentence was deleted when rewriting the discussion to be more concise.

L267: Please correct typo “presume” to “resume”

Thank you, we corrected it.

L269: Advise changing “many” to “previous”

Thank you, we changed to previous.

L376: Please specify the range of exposure times.

We specified the exposure time now; it’s approximately 1 sec. See lines 385-386: “GVOs were stained for 15 min with 0.1 µg/ml Hoechst 33342 and manually sorted by exposure to UV light for approximately 1 sec using an inverted fluorescence microscope”.

L514: Figure 3 legend should state that the fluorescence signal is pseudocoloured in the merged panels.

We thank the reviewer for the comment. We added this to the figure legend.

Reviewer #2 (Remarks to the Author):

In this paper, Eleftheriou et al. asked if germinal vesicle oocytes (GVOs) with decondensed chromatin (non-surrounded-nucleolus, NSN) transition to GVOs with dense perinuclear chromatin (surrounded nucleolus, SN). This is an interesting and important question in the field. Authors have examined epigenetic, morphological, and cytoplasmic changes in the NSN- to SN-GVO transition. Furthermore, DNA methylation was examined in parthenogenetic (PA) embryos derived from NSN-GVOs and SN-GVOs, in order to see whether the germ cell epigenome is transmitted to the next generation. These analyses indicate that SN-GVOs, intermediate-type GVOs (INT-GVOs), and NSN-GVOs show distinct patterns of epigenetic, morphological, and cytoplasmic states. Especially, DNA modification patterns are variable depending on each oocyte stage, and the methylation levels

in oocytes affect DNA methylation in 1-cell PA embryos. This paper confirms the previous findings regarding the differential nuclear and cytoplasmic states between NSN-GVOs and SN-GVOs, further extending their characterization to INT-GVOs. Authors have claimed that NSN-GVOs transition via INT-GVOs (intermediate stage) to SN-GVOs ex vivo. However, this main conclusion is not sufficiently supported by the presented data in this manuscript, as summarized below in this referee's comments. In addition, some of the conclusions are not well supported by the presented data.

Major concerns

1. Authors have claimed that murine NSN-GVOs transition via an intermediate stage into SN-GVOs. This important conclusion is mainly based on the observation that INT-GVOs and SN-GVOs are observed after the culture of NSN-GVOs under the "maturation inhibiting condition". Importantly, NSN-GVOs were selected by Hoechst 33342 staining (Methods), which has been shown to be toxic for development (Versieren, K. et al., 2014 *Zygote*, 22, 32-40). Therefore, these experiments were carried out in a non-physiological condition, the main conclusion has to be confirmed by another experiment that should better represent a physiologically relevant condition. One suggestion is to trace the transition between NSN-GVOs to SN-GVOs by live cell imaging. Authors have indeed tried a similar experiment in Figure 4C, but unfortunately the resolution of the images is not sufficient to claim the transition from NSN-GVOs to SN-GVOs. Furthermore, the initial classification method for NSN-GVOs, INT-GVOs, and SN-GVOs is extremely important for this paper but is not well explained in the current manuscript.

We thank the reviewer for these suggestions and comments. Concerning the non-physiological condition (potential toxicity of Hoechst 33342), we want to point out that in the paper cited by the reviewer the experimental setup was different, including staining with Hoechst 33342 after activation, with a concentration that is 5-10x higher than the one we used and an exposure time of 10 sec, where we keep this low at approximately 1 sec. Nevertheless, we understand the concerns raised by the reviewer, as actually, when we started working with GVOs, we were also concerned about Hoechst 33342 staining and UV exposure. However, when we perform experiments on Hoechst-stained SN-GVOs, those develop at normal rates until the blastocyst stage (after IVM and IVF) (see⁶). Moreover, also other studies show that sorting GVOs by Hoechst staining using low concentrations is not harmful to early development^{2,7,8}. Together, we are convinced that our conditions are physiological and that these experiments are valid to

determine the timing of our observed *ex vivo* transition. Although, of course, we agree that these conditions should never be used to discriminate between SN- and NSN-GVOs in reproductive medicine, neither will the injection of H2B-YFP mRNA.

As also mentioned by the other reviewers, we missed clearly defining the three states. We now define the GVO types more clearly and thank the reviewer for pointing this out. For more details see the answer to reviewer #2 comment 2 below.

2. The definition of INT-GVOs is vague. Authors need to clearly explain the criteria for classifying NSN-GVOs, INT-GVOs, and SN-GVOs. Quantitative statements are ideal for this classification. The difference between NSN-GVOs and INT-GVOs is not clear from Figure 1A.

We thank the reviewer for this comment. We have now more clearly defined the different types of GVOs and chose better quality images and magnifications for Fig. 1a. We added this to the Method section: (lines 386-391) “GVOs, which show diffuse Hoechst staining in the nucleus accompanied by a few dot-like stained denser regions,

were categorized as NSN-GVOs; GVOs, which show a partial perinucleolar ring-like structure, as INT-GVOs, and GVOs that show a fully closed perinucleolar ring of Hoechst staining in the nucleus as SN-GVOs (for examples see Fig. 1a). GVOs, which could not be clearly categorized, were excluded from further experiments.”

We also agree that a non-biased quantitative approach to characterize NSN- and SN-GVOs would be the best, but, unfortunately, is not feasible for experiments where further culturing of the oocytes is needed due to minimizing exposure to UV light. Hoechst staining to visualize the perinucleolar heterochromatin is so far the gold-standard method in the field and we want to note that NSN-, INT-, and SN-GVOs can be easily distinguished using a Hoechst staining. GVOs, which are borderline between the different states and could not be clearly categorized were excluded from further analysis.

We also compared our criteria with published studies, all using similar definitions:

- Bonnet-Garnier and colleagues propose that in SN-oocytes, the chromatin is compacted and forms a ring around the nucleolus whereas in the NSN-state the chromatin is not condensed. INT-oocytes are defined as GVOs that show less condensed chromatin and a partial ring of chromatin around the nucleolus (see³).
- Zuccotti and colleagues define NSN-GVOs as oocytes with a Hoechst-positive chromatin pattern of small clumps forming a network lining much of the nuclear surface, and SN-GVOs having a thread-like organization, surrounding the nucleolus. Further, they characterize INT-GVOs as oocytes between the NSN- and SN-configuration with a nuclear structure similar to the NSN configurations but with 2-3 patches of perinucleolar chromatin clearly visible (see⁴).
- Lin and colleagues define NSN-GVOs as oocytes with diffuse chromatin which does not form a heterochromatin rim around the nucleolus and SN-GVOs with condensed chromatin that is particularly confined around the nucleolus, with almost no diffuse chromatin (see⁹).

Moreover, we want to mention that by performing a more detailed analysis of *ex vivo* derived N-SN-GVOs (see also reviewer 3 comment 3), we observed two types of SN-GVOs, which we

now also mention in the method part and in the main text. We added the following text in the method section (see lines 391-394): “NSN- *ex vivo* derived SN-GVOs (N-SN-GVOs) were subcategorized into type I and type II, with type I showing a diffuse Hoechst signal all over the nucleoplasm next to the dense perinucleolar ring-like structure, while type II is showing a dense perinucleolar Hoechst signal (Supplementary Fig. S6a).”

3. When the developmental competence of NSN-GVOs and SN-GVOs was examined (Figure 4a and 4b), how did you select those oocytes with different categories before being subjected to development tests? Did you use Hoechst 33342 staining? This needs to be explained in the result section.

Yes, we used Hoechst staining and we describe this now more clearly in the result section (lines 226-227): “To this end, we first characterized *in vitro* maturation and parthenogenetic activation efficiencies from pre-sorted (by Hoechst 33342 staining) NSN- and SN-GVOs.”

4. Live cell imaging is a good way forward to answer the main question of this paper. However, the images shown in Figure 4c do not represent the sufficient quality. For example, it is not possible for readers to distinguish NSN-GVOs and SN-GVOs from Figure 4c. Improvement for the live cell experiments is required. Non-biased quantitative approach to assess NSN-GVOs and SN-GVOs would be preferable.

We agree that the resolution of the images from the time-lapse experiments is not ideal, thus

we took better images and/or present the GFP-visualized nucleus now in higher magnification (updated Fig. 4c). We also marked the location of the nucleolus with an asterisk and hope that the reviewer now agrees with us that it is visible that NSN-GVOs perform GVBD before previous transitioning to a ring-like DNA staining (see also comment 3 from reviewer #3).

For the time-lapse imaging, we also want to clarify that before injecting H2B-YFP mRNA, we first performed a Hoechst

staining to be sure to start with an NSN-GVO – since the GVO type is better visible using direct DNA staining and we added this to the text (lines 233-234: “...we microinjected Hoechst-presorted NSN-GVOs or SN-GVOs with mRNA coding for histone H2B-YFP...”). We also agree that a non-biased approach to characterize NSN- and SN-GVOs would be the best; however, in this setup, it is rather unfeasible (as discussed in reviewer #3 comment 2).

5. Some of the conclusions have to be toned down. For example, to conclude “Our study provides novel insight into the dynamic changes of Tet enzyme-mediated DNA modifications (Line315-316)”, knockdown or knockout experiment of Tet in oocytes needs to be performed. To conclude “When this transition is compromised during late oogenesis, the developing preimplantation embryo is characterized by an abnormal epigenetic profile coinciding with developmental arrest” (Line357-359), you need to do more comprehensive studies including fertilized embryos using the NSN-GVOs and SN-GVOs.

We thank the reviewer for the valuable feedback and agree that more experiments would be needed to justify some of our findings. We toned down our conclusions accordingly and now state (line 339-341): “The changes of DNA modifications observed in this study during the NSN-SN transition reveal an additional phase of DNA methylation remodeling in late oogenesis in the transition to transcriptional silence.” Previous lines 357-359 are not anymore present in our more concisely written discussion.

6. Regarding the statistical tests with 3 samples, authors have used one-way ANOVA. One-way ANOVA detects significant differences among samples, but not each sample combination. However, P values are shown in each comparison. Can you check and clarify which statistical tests (Multiple Comparison Procedure) were used?

We thank the reviewer for noting this. We have now provided more explanation for the statistical methods used for this study, clarifying which multiple comparison procedure was used in the Methods section “Statistics and reproducibility” (see lines 488-495: “Statistical analyses were performed using GraphPad Prism software v9 (San Diego, CA USA). A value of $p < 0.05$ was considered statistically significant. Unless otherwise stated in the figure legend, data are shown as the mean \pm standard deviation, Student’s t-test (unpaired) was used for comparisons between two groups, and for multiple comparisons, one-way ANOVA followed by post-hoc Tukey’s multiple comparison test was used. The nature of the statistical tests and the p-values for the statistical analyses are provided in the associated figure legends wherever applicable. Reproducibility of results was ensured through biological replicates and/or independent repetitions, which have been explicitly mentioned wherever applicable.”).

Minor points

7. Line155-173, different types of mitochondria were observed in the different GVO types without strong correlation (Figure 2e). Can you explain more about this in the discussion section?

We discussed with electron microscopy experts the correlation between the structure and function of mitochondria, and also to our surprise, this issue has not been solved to our best knowledge. We carefully searched the last 60 years of electron microscopy studies about mitochondria and besides correlations of different cell types, this has not been addressed so far. We now added a sentence describing this in lines 164-166: "Even though different types of mitochondria are observed in the different GVO types, the correlation between morphology and activity of mitochondria is still not well understood and deserves future investigations."

8. Line458, more explanations are required for statistical methods used for this study.

We now added more details to this section (see also reviewer #2 comment 6).

Reviewer #3 (Remarks to the Author):

The manuscript, "Dynamic changes of DNA modifications in the transition phase of mouse germinal vesicle oocytes impact DNA methylation reprogramming in the early embryo" by Eleftheriou et al compared the differences in ultrastructure, maturation efficiency and epigenetic modification of oocytes with two different nuclear structures and proved the existence of an intermediate state between NSN- and SN- state. This is an interesting study that advances the understanding of oogenesis, but there are still some questions to be addressed.

1. Oogenesis takes place in the follicle, but oocyte growth is not synchronized with folliculogenesis. Therefore, it is important to clarify which types of follicles (Type 4, Type5, Type 6, or Type 7-8 follicle, refer to: Pedersen T and Peters H., Reproduction, 1968; Chen Y et al., Mol Hum Reprod, 2020) NSN-GVOs and SN-GVOs exist respectively? At what stage of folliculogenesis does NSN/SN transition occur?

We carefully compared all information we gathered on NSN- and SN-GVOs and from previously published studies^{2,4,10-12} with the follicular stages described in Pederson *et al.* 1968¹³ and now refer in the introduction to GVOs in Type 7-8 follicles (see lines 42-43):

“Interestingly, when GVOs are isolated from ovaries from type 7-8 follicles (see¹³), they can be classified into different states by their nuclear architecture.”

2. In this study, oocytes were collected from ovaries 48h after PMSG injection for analysis. Does PMSG participate in the regulation of NSN to SN transition? In other words, whether the proportion of NSN-, INT- and SN-GVOs in the population of oocytes changes before and after PMSG injection. This problem can be elucidated by superstimulation in juvenile mice (Postnatal Days 19-20).

The reviewer poses an interesting question here. A previous report has shown that in juvenile mice (4-6 weeks old) the SN-GVO population increases in injected females compared to the non-injected⁴, which we also observed in adult mice (>8 weeks old; new Supplementary Fig. S8a). To further address this question, as suggested by the reviewer, we examined the impact of PMSG on pre-pubertal mice (postnatal day 14). Here, in both conditions, with and without PMSG, we were exclusively able to harvest NSN-GVOs. We also checked if those GVOs can transition *in vitro* and while ~27% start transitioning, we did not observe a difference when PMSG was injected. We also analyzed PMSG injected 17 days old mice and isolated GVOs on day 19, and made similar observations as in adult mice (new Supplementary Fig. S8a, b). Thus, we exclude an effect of PMSG on our observations concerning the *ex vivo* NSN-SN transition.

We added these experiments as new Supplementary Fig. S8 and text in the method section in the lines 396-401: “PMSG injection was used for synchronizing the GVOs. Injection of PMSG leads to a slight increase of SN-GVOs and a decrease of NSN-GVOs in adult mice and juvenile mice on day 19 (Supplementary Fig. S8a). In young mice, before the onset of natural SN transitions, the injection of PMSG increases the number of GVOs that can be obtained without inducing the NSN-SN transition (Supplementary Fig. S8a, b). Upon *ex vivo* transition, there is no difference in the transition rate to the SN-state in juvenile mice with or without PMSG injection (Supplementary Fig. S8c).”

3. Fig. 1F showed that NSN-GVOs can be converted to SN-GVOs during in vitro culture. Is the SN-GVOs converted from NSN-GVOs the same as the natural SN-GVOs? That is, do the organelles of the two types of SN-GVOs follow a similar distribution? Do they still differ in size? Are they equally efficient at nuclear maturation? Do they have similar epigenetic modifications and the same potential to develop after fertilization? These questions will determine the practical value of the study and are also of interest to readers.

We thank the reviewer for this indeed interesting question, which we agree will add more value to the study towards direct practical use in reproductive medicine.

To analyze how successful this *ex vivo* observed transition compared to naturally obtained SN-GVOs is, we performed a new set of experiments on *ex vivo* transitioned GVOs. We included these new data as new Supplementary Fig. S6-7 and added new text at lines 268-308. To shortly summarize: Using Hoechst staining we observed two types of *ex vivo* NSN-derived-SN-GVOs (hereby mentioned as N-SN-GVOs), one with diffuse chromatin all around the nucleus together with a ring of heterochromatin around that nucleolus (type I) and one with a staining pattern resembling the *in vivo* SN-GVOs (type II). The fraction of type II N-SN-GVOs increases over time during this transition. Regarding oocyte volume, mitochondria localization, and 5mC, we show that the *ex vivo* transitioned N-SN-GVOs share these characteristics with the SN-state or show a tendency towards the SN-state characteristics (5caC modifications). However, while we observed a higher fertilization rate in N-SN-GVOs than in NSN-GVOs, these oocytes did not develop to blastocysts and arrested at the 2-cell stage. Likely, conditions for the *ex vivo* transition need to be optimized – e.g., timing or media composition, which should be the aim

of future studies. Importantly, also SN-GVOs which were cultured for 48 h under the same conditions were showing a 2-cell arrest, which let us speculate that the transcriptional quiescent state can only be maintained for a specific time to still support developmental competence.

4. In my opinion, Fig. 4C fails to distinguish the difference in nuclear structure between NSN- and SN-GVOs. What's more, it is difficult to find differences in nuclear maturation between NSN- and SN-GVOs in the movies presented by the authors. If differences exist, then the authors need to make detailed notes for the reader to understand.

We agree with the reviewer that the images from the time-lapse experiments are not having the highest resolution. This is due to the time-lapse microscope and the dishes we use to culture the oocytes. While this setup is great for brightfield images, fluorescence images are not as high of a quality as they would be on a confocal microscope. We have to note that extensive

exposure to fluorescence will harm oocytes/embryos, so we keep the exposure time to a minimum. Importantly, we want to clarify that in these experiments we sorted the oocytes first by Hoechst staining before injection so that we can be sure that we start with an NSN- or SN-GVO (lines 233-234: “..we microinjected Hoechst-presorted NSN-GVOs or SN-GVOs with mRNA coding for histone H2B-YFP..”). We now updated Fig. 4c and provide a new Supplementary Movie 2 for the SN-GVO. We are convinced that the types can be

clearly discriminated in these new movies/images. In the Figures, we also marked the location of the nucleolus with an asterisk for easier recognition of the nucleolus, where the pericentromeric nucleolar ring should be found and show a magnification of this area.

5. During the transition from NSN- to SN-GVOs, the morphology and distribution of mitochondria changed significantly. It is suggested that the authors further examine the

changes of mtDNA and ATP. Otherwise, the descriptions in line141-143 and line153-154 would lack sufficient support.

We thank the reviewer for this comment. We realized that we did not appropriately describe in the mentioned sentences on which data our description is based. We changed lines 141-143 (now lines 143-146) to: “We observed a significant increase in mitochondrial mass in the transition from NSN- to SN-GVOs through the MitoTracker Green staining (Fig. 2b, c) accompanied by a proportional increase in the mitochondrial activity (Fig. 2b, Supplementary Fig. S2b), with INT-GVOs showing similar characteristics as NSN-GVOs.”. We also talked to experts in the field studying mitochondria and they agreed with us that TMRM staining is appropriate to analyze mitochondrial activity. As stated above, we also reformulated our description – since we could imagine that it could have been misunderstood, that we want to state that mitochondrial activity increases per mitochondria – what we mean is that mitochondrial activity per mitochondria stays the same, just the number of mitochondria increases, as visible in the high correlation coefficient in Fig. 2b.

Furthermore, we deleted previous lines 153-154, since we agree that we actually don't directly show mitochondrial proliferation.

Analyzing changes in mtDNA content is another way of showing an increase in mitochondrial mass next to our approach using MitoTracker staining. We also attempted to quantify mitochondria by performing qPCR on mitochondrial DNA in NSN- and SN-GVOs. As seen in Reviewer Fig. R1, we also see a tendency for more mtDNA in SN-GVOs in the qPCR-results (lower Ct-values). However, this is not statistically significant, which is likely due to high technical variations and in general only a small increase in mitochondrial mass (MitoTracker Green IF-signal increases 1.3-fold). Thus, we decided to not include this figure in the manuscript.

OTHER CONCERNS

1. The ABSTRACT is not enough to summarize the research content of the manuscript, please rewrite it.

We have rewritten the abstract to better summarize the research content.

2. Line 136: Please use volume to measure the change in oocyte size.

We thank the reviewer for this comment and updated the figure and now show volume instead of diameter (see updated Fig. 2a).

3. Line 185: It is not recommended to use "data not shown" to make a vague statement.

We thank the reviewer and totally agree that "data not shown" should not be used in manuscripts. We compiled a set of images to show this now and added a new Supplementary Fig. S3.

4. Line 155-159: No data is presented for the results described

The general distribution of mitochondria is hard to show due to the high magnification used to visualize mitochondria in our TEM images. Thus, we show the distribution in images in the paper by the MitoTracker Green staining (Fig. 2d) and now we removed these lines from the main text.

5. Line 161: The authors classify mitochondria into different types and describe the structural characteristics of each type in detail. This is great. However, I believe that readers would be more interested in the functional differences of different mitochondrial types, so I suggest the authors ask experts in mitochondrial research for advice and add descriptions about the functional differences between different mitochondrial types in the manuscript. Similarly, the description of different types of Golgi apparatus should be the same.

We performed an extensive literature research and talked to several TEM or experts studying mitochondria on what is known about the function of mitochondria in relation to their phenotypic appearance in TEM images, but unfortunately to our best knowledge, this information is missing so far from the mitochondria research field, which is also the case for the Golgi apparatus. At this point, we can only provide speculations and future experiments should address these points.

We added lines 165-166: "...the correlation between morphology and activity of mitochondria is still not well understood and deserves future investigations."

6. Fig.4c: The perivitelline space of oocytes in the SN-GVOs group was abnormally large.

We agree that the SN-GVO we chose shows an abnormally large perivitelline space. Now we provide a better representative example of an SN-GVO with better and higher quality images.

7. The DISCUSSION is lengthy and does not succinctly highlight the innovation and importance of the manuscript. In addition, the discussion on epigenetic modification is too detailed, but there is no in-depth study on epigenetic modification in the RESULTS.

We thank the reviewer for pointing this out and now provide a much more concise discussion and more relevantly highlight our findings.

References

- 1 Inoue, A., Nakajima, R., Nagata, M. & Aoki, F. Contribution of the oocyte nucleus and cytoplasm to the determination of meiotic and developmental competence in mice. *Hum Reprod* 23, 1377-1384, doi:10.1093/humrep/den096 (2008).
- 2 Debey, P. et al. Competent mouse oocytes isolated from antral follicles exhibit different chromatin organization and follow different maturation dynamics. *Mol Reprod Dev* 36, 59-74, doi:10.1002/mrd.1080360110 (1993).
- 3 Bonnet-Garnier, A. et al. Genome organization and epigenetic marks in mouse germinal vesicle oocytes. *Int J Dev Biol* 56, 877-887, doi:10.1387/ijdb.120149ab (2012).

- 4 Zuccotti, M., Piccinelli, A., Giorgi Rossi, P., Garagna, S. & Redi, C. A. Chromatin organization during mouse oocyte growth. *Mol Reprod Dev* 41, 479-485, doi:10.1002/mrd.1080410410 (1995).
- 5 Correll, C. C., Bartek, J. & Dunder, M. The Nucleolus: A Multiphase Condensate Balancing Ribosome Synthesis and Translational Capacity in Health, Aging and Ribosomopathies. *Cells* 8, doi:10.3390/cells8080869 (2019).
- 6 Arand, J. et al. Tet enzymes are essential for early embryogenesis and completion of embryonic genome activation. *EMBO Rep* 23, e53968, doi:10.15252/embr.202153968 (2022).
- 7 Zuccotti, M. et al. Meiotic and developmental competence of mouse antral oocytes. *Biol Reprod* 58, 700-704, doi:10.1095/biolreprod58.3.700 (1998).
- 8 Zuccotti, M. et al. The analysis of chromatin organisation allows selection of mouse antral oocytes competent for development to blastocyst. *Zygote* 10, 73-78, doi:10.1017/s0967199402002101 (2002).
- 9 Lin, J. et al. The relationship between apoptosis, chromatin configuration, histone modification and competence of oocytes: A study using the mouse ovary-holding stress model. *Sci Rep* 6, 28347, doi:10.1038/srep28347 (2016).
- 10 Bouniol-Baly, C. et al. Differential transcriptional activity associated with chromatin configuration in fully grown mouse germinal vesicle oocytes. *Biol Reprod* 60, 580-587 (1999).
- 11 Mattson, B. A. & Albertini, D. F. Oogenesis: chromatin and microtubule dynamics during meiotic prophase. *Mol Reprod Dev* 25, 374-383, doi:10.1002/mrd.1080250411 (1990).
- 12 Wickramasinghe, D., Ebert, K. M. & Albertini, D. F. Meiotic competence acquisition is associated with the appearance of M-phase characteristics in growing mouse oocytes. *Dev Biol* 143, 162-172, doi:10.1016/0012-1606(91)90063-9 (1991).
- 13 Pedersen, T. & Peters, H. Proposal for a classification of oocytes and follicles in the mouse ovary. *J Reprod Fertil* 17, 555-557, doi:10.1530/jrf.0.0170555 (1968).

REVIEWERS' COMMENTS:

Reviewer #1 (Remarks to the Author):

The authors have conducted additional experiments and revised their manuscript to incorporate satisfactory changes in response to this reviewer (and in my opinion, to each of the other reviewers). Indeed, in my view, the authors' response was exemplary (as indicated in the detailed rebuttal and the revised, Tracked Changes manuscript and supplementary information). Accordingly, I compliment the authors for their novel findings reported in the manuscript, which add significantly to our understanding of the biology and epigenetic mechanisms underlying the acquisition of oocyte developmental competence.

Reviewer #2 (Remarks to the Author):

The authors have satisfactorily addressed my concerns in the revised manuscript,

Reviewer #3 (Remarks to the Author):

I have reread the manuscript, and the author addressed all my concerns with experiments. Compared with the previous edition, the manuscript has been. I have no more questions